# The Atlantic's Freshwater Budget under Climate Change in the Community Earth System Model with Strongly Eddying Oceans

André Jüling[1], Xun Zhang[1], Daniele Castellana[1], Anna S. von der Heydt[1], and Henk A. Dijkstra[1]

[1]Institute for Marine and Atmospheric research Utrecht (IMAU), Utrecht University, Netherlands

**Correspondence:** André Jüling (a.juling@uu.nl)

**Abstract.** We investigate the freshwater budget of the Atlantic and Arctic oceans in coupled climate change simulations with the Community Earth System Model and compare a strongly eddying setup with $0.1°$ ocean grid spacing to a non-eddying $1°$ configuration typical of CMIP6 models. Details of this budget are important to understand the evolution of the Atlantic Meridional Overturning Circulation (AMOC) under climate change. We find that the slowdown of the AMOC in the year 2100 under the increasing $CO_2$ concentrations of the RCP8.5 scenario is almost identical between both simulations. Also, the surface freshwater fluxes are similar in their mean and trend under climate change in both simulations. While the basin-scale total freshwater transport is similar between the simulations, significant local differences exist. The high ocean resolution simulation exhibits significantly reduced ocean state biases, notably in the salt distribution, due to an improved circulation. Mesoscale eddies contribute considerably to the freshwater and salt transport, in particular at the boundaries of the subtropical and subpolar gyres. Both simulations start in the single equilibrium AMOC regime according to a commonly used AMOC stability indicator and evolve towards the multiple equilibrium regime under climate change, but only the high resolution simulation enters it due to the reduced biases in the freshwater budget.

**Keywords.** Atlantic Meridional Overturning Circulation

Mesoscale ocean flows

Climate Change

Ocean freshwater transports

Freshwater budget

# 1 Introduction

One of the important Tipping Elements in the climate system (Lenton et al., 2008) is the Atlantic Meridional Overturning Circulation (AMOC). This component of the ocean circulation carries about $1.5\,\mathrm{PW}$ of heat northwards at 26.5°N in the Atlantic (Johns et al., 2011) and hence its strength and spatial expression significantly affect local surface temperature and precipitation (Palter, 2015). The potential tipping character of the AMOC is expressed through large and abrupt changes in AMOC strength (Srokosz et al., 2012; Weijer et al., 2019), for which evidence exists in the palaeo record (Lynch-Stieglitz, 2017). In models of the AMOC, such transitions can occur due to the existence of multiple equilibria, where several AMOC states can coexist under the same forcing conditions. Such multiple equilibria of the AMOC have been found in a hierarchy of ocean-climate models (Stommel, 1961; Rahmstorf et al., 2005; Hawkins et al., 2011; Toom et al., 2012; Mecking et al., 2016). They occur due to the presence of positive feedbacks, the most prominent one being the salt-advection feedback (Peltier and Vettoretti, 2014). Subtle changes to the freshwater budget can modify the AMOC response to perturbations which is why the correct simulation of the oceanic freshwater budget in the Arctic and Atlantic is important (Behrens et al., 2013).

The Atlantic is a net evaporative basin resulting in the saltiest subtropical surface waters of all the major oceans. At the Atlantic's southern boundary, which we take to coincide with the southern tip of Africa at 34°S, relatively salty surface waters together with fresh Antarctic Intermediate Waters are imported in the upper $1000\,\mathrm{m}$. This northward transport amounts to approximately $17\,\mathrm{Sv}$ at 26.5°N (Moat et al., 2018; Smeed et al., 2018; Frajka-Williams et al., 2019). At high northern latitudes the surface waters are transformed into North Atlantic Deep Water (NADW) which returns southwards and is exported at 34°S. A lower, weaker overturning cell exists in which cold Antarctic Bottom Water enters the South Atlantic at the bottom and returns just above with the NADW. Salt also enters the South Atlantic from the Southwest Indian Ocean via Agulhas leakage in the form of eddies shed off the Agulhas retroflection (McDonagh et al., 1999). From the North, approximately $0.8\,\mathrm{Sv}$ of relatively fresh Pacific water enters the Arctic Ocean via Bering Strait where it further freshens primarily due to river discharge from the large Arctic catchment area (Woodgate and Aagaard, 2005). Together with freshwater in the form of sea ice, relatively fresh seawater enters the Atlantic from the North. In the Strait of Gibraltar, relatively fresh surface waters flow into the strongly evaporative Mediterranean Sea and saltier waters return into the Atlantic at depth. The meridional asymmetry of the precipitation pattern of the Intertropical Convergence Zone (ITCZ) results in salinity differences between the North and South Atlantic. The wind driven subtropical and subpolar gyres recirculate water primarily horizontally and advect any zonal salinity gradient also in the meridional direction.

As atmospheric temperatures rise under increasing greenhouse gas concentrations, the hydrological cycle generally strengthens making dry regions drier and wet regions wetter, amplifying sea surface salinity patterns (Held and Soden, 2006; Skliris et al., 2020). The AMOC is projected to weaken under climate change due to buoyancy flux changes as heat flux and net precipitation patterns change (Stocker et al., 2013). The heat flux changes are the dominant driver of AMOC strength reduction and there is evidence that this slowdown is already underway (Gregory et al., 2005; Caesar et al., 2018). In order to judge how fast the AMOC can change and whether it could collapse abruptly, one needs to assess the AMOC stability and in particular the strength of the positive feedbacks. In CMIP5 models, no transition to a different statistical equilibrium state is found up

to the year 2100 under any of the climate change scenarios (Cheng et al., 2013) and it remains unclear whether the AMOC is already in or will shift into a multiple equilibrium regime, which would allow such transitions (Gent, 2018).

Many studies have linked the freshwater budget, through the salt-advection feedback, to the response of the AMOC under surface freshwater perturbations (Rahmstorf, 1996; de Vries, 2005; Dijkstra, 2007; Mecking et al., 2017; Liu et al., 2017). The existence of a multiple equilibrium regime is connected to the sign of the divergence of the advective AMOC induced Atlantic freshwater transport $\Sigma$ (or $\Delta M_{ov}$ in Liu et al. (2017)) which exactly marks the separation of the unique and multiple equilibrium regimes when atmospheric feedbacks are negligible (Dijkstra, 2007). As the northern boundary freshwater transport is minor this divergence is often approximated by its southern boundary component only, referred to as $M_{ov}$ (de Vries, 2005), $F_{ov}$ (Hawkins et al., 2011), or $F_{ovS}$ (Weijer et al., 2019). We will use $F_{ovS}$ here as we use $F$ to denote freshwater fluxes in general and $F_{ov}$ for the latitudinally dependend overturning component in particular. We define freshwater relative to a salinity of $S_0 = 35$ and detail the computations of the different transport components in Appendix B. Positive $F_{ovS}$ values indicate that the AMOC imports freshwater which constitutes a negative feedback as a positive AMOC strength perturbation would be damped by an enhanced freshwater import into the North Atlantic suppressing deep water formation. A negative $F_{ovS}$ value, on the other hand, would induce an amplification of an AMOC perturbation (Huisman et al., 2010). Observational estimates of $F_{ovS}$ are negative suggesting multiple AMOC equilibria in the present-day climate (Weijer et al., 2019). The models of the last two model intercomparison projects CMIP3 and CMIP5 tend to have positive $F_{ovS}$ values due to a salinity bias at 34°S where the upper water masses are too fresh and the deep southward return flow is too salty (Drijfhout et al., 2011; Weaver et al., 2012; Liu et al., 2014; Mecking et al., 2017). Once this bias is accounted for, $F_{ovS}$ values for most models lie within the range of observations (Mecking et al., 2017). Under increasing radiative forcing, CMIP3 models exhibit a negative $F_{ovS}$ trend (Drijfhout et al., 2011), but no consistent sign in this trend is found in CMIP5 models (Weaver et al., 2012).

Refining the grid spacing from $1°$ typical of CMIP5 and CMIP6 ocean model components to $0.1°$ resolves the internal Rossby radius of deformation over large parts of the ocean (Hallberg, 2013). This enables the development of eddies, filaments, and fronts through mixed barotropic/baroclinic instabilities and the simulation of other mesoscale ocean features such as currents at the western boundary and through narrow straits. We use the terminology 'strongly eddying' for ocean grids with $0.1°$ horizontal grid spacing as these are neither just eddy-permitting (typically $0.25°$) nor fully mesoscale turbulence resolving (Moreton et al., 2020). These high resolution ocean models constitute the only consistent method to estimate eddy contributions to ocean variability and the mean climate state and generally result in significantly reduced ocean biases (Kirtman et al., 2012; Small et al., 2014). The eddy freshwater transport is comparable in magnitude to the mean transport at the poleward and equatorward boundaries of the subtropical gyres (Treguier et al., 2012, 2014). Some of this transport will be captured by eddy parametrizations in low resolution simulation, but other effects, such as the advection of salt by Agulhas Rings, cannot be captured adequately.

Relatively few studies have investigated the AMOC behavior in strongly eddying ocean models (Weijer et al., 2012; den Toom et al., 2014; Brunnabend and Dijkstra, 2017; Hirschi et al., 2020). The improved simulation of overflows over sills in high resolution models significantly reduces deep water density biases which leads to improved simulation of deep convection. The pathway of the North Atlantic Current and the formation sites of North Atlantic Deep Water are more realistic at high

resolution (Hirschi et al., 2020). A comparison of the AMOC response between 10 GFDL models under a 1% per year $CO_2$ increase scenario showed that in coarse resolution models, the AMOC declines between 16% and 45%, and the eddy-permitting and strongly-eddying configurations are at the lower end of these percentages with 13% and 16%, respectively (Winton et al., 2014). The POP ocean model showed qualitatively similar AMOC responses to surface freshwater perturbations between strongly eddying ($0.1°$) and non-eddying ($1°$) model configurations (Weijer et al., 2012; den Toom et al., 2014; Brunnabend and Dijkstra, 2017), but with a dependence on the location of the perturbation. However, whether ocean model resolution affects the AMOC response to forcing systematically remains an open question (Gent, 2018), although there is evidence from eddy-permitting models that the AMOC mean state, in particular the sites of deep water formation, controls the response (Jackson et al., 2020). A suite of high resolution ocean-only hindcasts with the NEMO model at $1/12°$ show that the stability indicator $F_{ovS}$ is negative (Deshayes et al., 2013) in contrast to coarse resolution coupled models (Mecking et al., 2017). The NEMO model thus shows a reduced bias in $F_{ovS}$, but it remains unclear how much of the bias reduction is due to the use of restoring boundary conditions within the ocean-only setup and how much due to an improved mean state of the ocean circulation.

As coarse resolution models exhibit biases in their mean state and lack mesoscale processes, the simulated sensitivity to forcing may be inadequate and the strength of the salt advection feedback may be affected. We investigate the effect of improving the ocean model resolution on the Atlantic freshwater budget and its sensitivity by analyzing present day control and high $CO_2$ concentration pathway simulations in two configurations of the Community Earth System Model, one with an ocean model grid spacing of $0.1°$ and the other with $1°$. The following section 2 describes these model simulations and provides a model-observation comparison of the control simulations. Section 3 presents the results, including changes to the AMOC, the Atlantic freshwater and salt budgets, and the effects on AMOC stability, under the climate change scenario. The results are summarized and discussed in section 4.

## 2 Model simulations and model-observation comparison

### 2.1 CESM simulations

We analyze four simulations with the Community Earth System Model version 1 (CESM, Hurrell et al. (2013)), carried out at the Academic Computing Center in Amsterdam (SURFsara), see e.g. van Westen and Dijkstra (2017). The CESM components are CAM5 (Community Atmosphere Model), POP2 (Parallel Ocean Program), CICE (sea ice model), and CLM (Community Land Model) which are coupled by the CESM1 coupler. The ocean model formulation is volume conserving and surface freshwater fluxes are thus modeled as virtual salt fluxes. The high resolution ('HR') simulation was performed with a $0.1°$ ocean horizontal grid spacing on a tripolar grid, while the low resolution ('LR') simulation was conducted with $1°$ ocean horizontal grid spacing with a displaced dipole grid. Tracer diffusing subgrid-scale processes are parameterized by the Gent-McWilliams scheme (Gent and McWilliams, 1990) in the $1°$ resolution simulation and by biharmonic diffusion in the $0.1°$ resolution case, which is strongly eddying.

Both control simulations use constant year 2000 atmospheric greenhouse gas concentrations forcing (notably $[CO_2] = 367\,\mathrm{ppm}$, $[CH_4] = 1760\,\mathrm{ppb}$). The HR-CESM simulation continues from an NCAR simulation of several decades which it-

120 self was initialized from a motionless ocean with a present-day estimate of the ocean's temperature and salinity distribution. The LR-CESM simulation was similarly continued from an NCAR provided initial state. The climate change simulations are following the $CO_2$ concentration of the highest Representative Concentration Pathway (RCP8.5) of the Coupled Model Intercomparison Project phase 5 (CMIP5) used in the Fifth Assessment Report of the Intergovernmental Panel on Climate Change (Stocker et al., 2013), but do not include other greenhouse gas increases or land use changes. We name the present-day control

simulation CTRL and the climate change simulation RCP. In 2100, the radiative forcing of $CO_2$ alone is $6.9\,\mathrm{W\,m^{-2}}$, or 80% of the $8.5\,\mathrm{W\,m^{-2}}$ of the RCP8.5 scenario (van Vuuren et al., 2011). Not prescribing land use changes, has no effect on the global mean surface temperaturature in the RCP8.5 scenario (Davies-Barnard et al., 2014). Compared to the mean warming in 2100 of the two RCP8.5 CESM1/CAM5 simulations submitted to CMIP5 at $4.4\,°\mathrm{C}$ (Meehl et al. (2013); time series available at https://climexp.knmi.nl/CMIP5/Tglobal/), our LR-CESM RCP simulation warmed only $2.9\,°\mathrm{C}$, or 66% of the RCP8.5 value.

The reduced warming until 2100 is both because of the aforementioned reduced radiative forcing, but also the fact that our simulation started from a nearly equilibrated, and hence relatively warm, year 2000 control simulation. The main characteristics of the model simulations are summarized in Table 1.

**Table 1.** Overview of the CESM simulations used, their ocean and atmosphere grid, the model version, as well as the year at which the RCP simulations are branched off the CTRL simulations.

| setup | ocean grid | | atmosphere grid | | CESM version | begin RCP [year] |
|---|---|---|---|---|---|---|
| HR-CESM | $0.1°$ tripole, 42 levels to $6000\,\mathrm{m}$ | `tx0.1v2` | $0.47° \times 0.63°$ | `f05` | 1.04 | 200 |
| LR-CESM | $1°$ dipole, 60 levels to $5500\,\mathrm{m}$ | `gx1v6` | $0.9° \times 1.25°$ | `f09` | 1.12 | 500 |

There are additional differences between the model configurations apart from the horizontal ocean model resolution. The $0.1°$ POP2 model grid has 42 levels to $6000\,\mathrm{m}$ while the LR-POP2 grid has 60 levels to $5500\,\mathrm{m}$. In contrast to the HR-

135 CESM ocean grid with its partial bottom cells and explicitly resolved overflows, the LR-CESM grid is defined with complete bottom cells and uses overflow parametrizations, e.g. between the Nordic Seas and the Atlantic (Smith et al., 2010). In the $0.1°$ POP2 model, the explicitly modeled Nordic Seas overflows compare favourably to observations (Ypma et al., 2019). The Mediterranean Outflow is not parameterized in the $1°$ POP2 grid but is modeled with a widened Strait of Gibraltar. Ultimately, the effect of the different vertical resolutions is hard to disentangle as the horizontal mixing is represented very

differently. Further, the CESM versions and atmosphere resolution differ between the HR-CESM (version 1.04) and the LR-CESM simulation (version 1.12). The newer version employs a different dynamical core in the atmosphere model (CAM5.2 versus CAM5.0) and some parameterization schemes are updated. In contrast to the improvement in ocean model resolution however, halving the atmospheric grid spacing from $1°$ to $0.5°$ is not resolving new essential physical processes. Therefore, no significant changes are expected between the $0.5°$ CAM5.0 HR-CESM and $1°$ CAM5.2 LR-CESM simulations' atmospheres

due their resolved physics apart from coupling to different ocean boundary conditions.

## 2.2 Model-observation comparison

To assess the performance of the HR- and LR-CESM CTRL simulations we use several observational datasets which are relevant for the freshwater budget and compare 30-year means of the CTRL simulations following the RCP branch-off point (years 200-229 and 500-529, see Tbl. 1). In many aspects the HR-CESM CTRL simulation performs better than the LR-CESM CTRL simulation when compared to the present-day climate. Global maps of the quantities presented here for the Atlantic-Arctic are included in Appendix A. When linear fits are presented, such as in Fig. 1d/e, significance of the fit is tested with a Wald test against a zero-slope null hypothesis.

We define regions in the Atlantic that approximately correspond to the subtropical gyres (STG; sometimes specified as South or North Atlantic: SA-STG and NA-STG), the subpolar gyre (SPG), the Intertropical Convergence Zone (ITCZ), and the Arctic. Green lines in Fig. 1b/c show bounding latitudes which are at the southern end of the Atlantic basin around 34°S, 10°S and 10°N generously bounding the ITCZ, 45°N as the approximate boundary of the subtropical and subpolar gyres, and 60°N as the boundary between the Atlantic and the Arctic. The Arctic Ocean includes Hudson Bay and is bounded on the Pacific side by the Bering Strait at 68°N. We perform the calculations on the original model 0.1° and 1° grids which become distorted relative to a regular latitude-longitude grid at high northern latitudes (see 60°N line in Fig. 1).

### 2.2.1 Sea surface temperature

The sea surface temperature (SST) is important for the freshwater budget as it strongly controls evaporation. Figure 1 shows the HadISST 1990-2019 SST climatology (Rayner et al., 2003), the bias of the CTRL simulations with respect to that climatology, and the linear SST trends of the RCP simulations. The HR-CESM (LR-CESM) simulation global mean SST is about $0.51\,(0.86)\,\mathrm{K}$ too warm with a RMSE of $0.99\,(1.39)\,\mathrm{K}$ compared to the HadISST dataset. Some warm bias is to be expected as the simulations are subjected to constant year 2000 radiative forcing and not the transiently increasing historical forcing. In the HR-CESM Atlantic, the sea surface is slightly too cold equatorward of 30° and too warm poleward of these latitudes with the exception of the South Atlantic near the African coast. The LR-CESM SST biases are stronger with warm biases in the South Atlantic, along the North American East Coast due to the Gulf Stream separating too far north, and north of 50°N. The LR-CESM simulation NA-STG and the southern edge of the NA-SPG are too cold resulting in asymmetric bias about the equator. Both simulations SSTs are too high in the NADW formation areas which results in warm biases in this water mass. The RCP SST trends are positive everywhere with a marked Arctic amplification. The exception is the NA-SPG with negative SST trends, an expected response associated with an AMOC decline under radiative forcing (Drijfhout et al., 2012).

### 2.2.2 Surface freshwater fluxes

We compare surface freshwater fluxes to ERA-Interim precipitation minus evaporation, $P-E$, 1989-2010 climatology (Dee et al., 2011; Trenberth et al., 2011). Figure 2 shows maps of the observed mean $P-E$ and the bias of the two simulations, and the zonally integrated $P-E$ fluxes. There is net evaporation in the STGs and net precipitation in the ITCZ just north of the equator (Fig. 2a) and the mid- to high-latitudes. Both simulations exhibit the same positive global precipitation biases

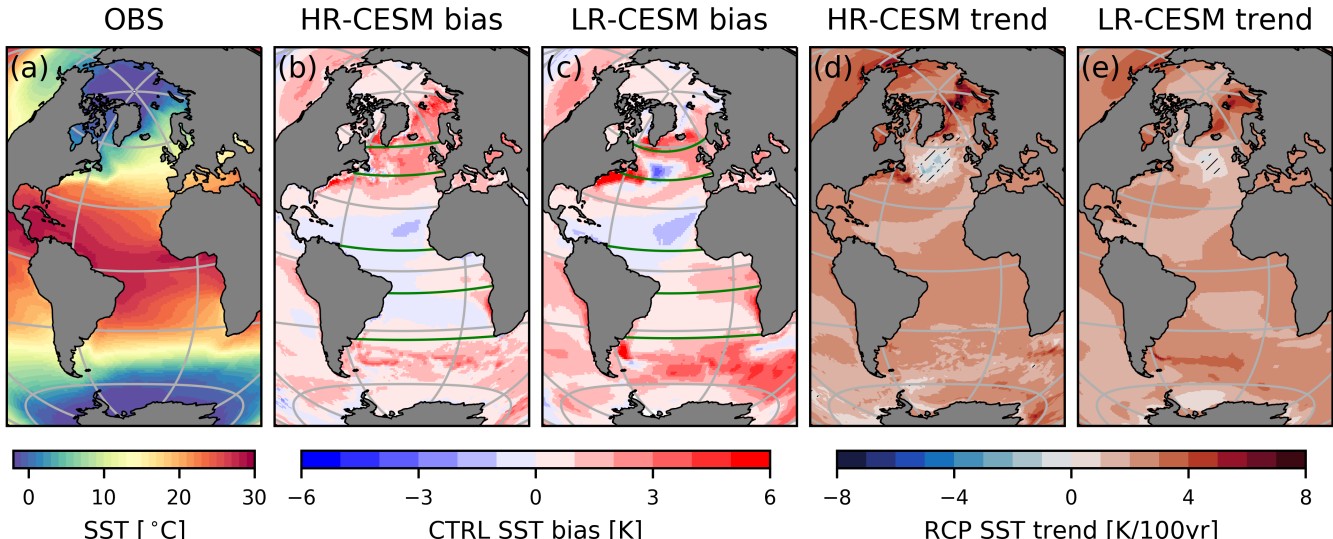

**Figure 1.** The sea surface temperature (SST) from the HadISST 1990-2019 observations (a), the SST bias of the HR-CESM (b) and LR-CESM (c) CTRL simulations, and the linear trends of the HR-CESM (d) and LR-CESM (e) RCP climate change scenarios. Hatched areas in the trend maps are not significant at the 5% level. The Lambert Azimuthal projection of these and subsequent maps is an equal area projection and grey parallels (meridians) are drawn every $30°$ $(60°)$. The green lines (b,c) show transects in the tripolar $0.1°$ and dipolar $1°$ POP2 model grid at 34°S, 10°S, 10°N, and approximately 45°N and 60°N. This northernmost meridional boundary differs from the 60°N parallel because of the curvilinear grids and is chosen to lie south of Hudson Strait.

of $0.23 \pm 1.01\,\mathrm{mm/day}$ (mean±RMSE; see appendix A). The $P - E$ bias is negative almost everywhere in the HR-CESM Atlantic (Fig. 2b) and over large parts of the LR-CESM Atlantic (Fig. 2c). Both simulations show biases around the ITZC, most noticeably with reduced precipitation near the South American coast north of the equator. The HR-CESM ITCZ appears slightly rotated with a wider precipitation belt in the central equatorial Atlantic and reduced precipitation in the Northwest and Southeast. The LR-CESM ITCZ is shifted south because the SST bias (cf. Fig. 1c) is meridionally asymmetric about the equator. As the surface waters diverge at the equator this contributes to the saline (fresh) surface bias of the North (South) Atlantic. Around the Gulf Stream too much water evaporates, but this is stronger and extends further north in the LR-CESM simulation, reflecting SST biases there (cf. Fig. 1c). Both the flux per degree latitude and the meridionally integrated flux referenced to zero transport at 34°S shown in Fig. 2d/e reveal comparable biases in the zonal integrals with minor differences in the ITCZ.

### 2.2.3 Salinity distribution

The heterogeneous salinity distribution in the ocean is the result of surface exchanges of freshwater and redistribution by the circulation. We use the EN4 global salinity observations averaged over 1990-2019 which is provided on a $1° \times 1°$ grid (Good

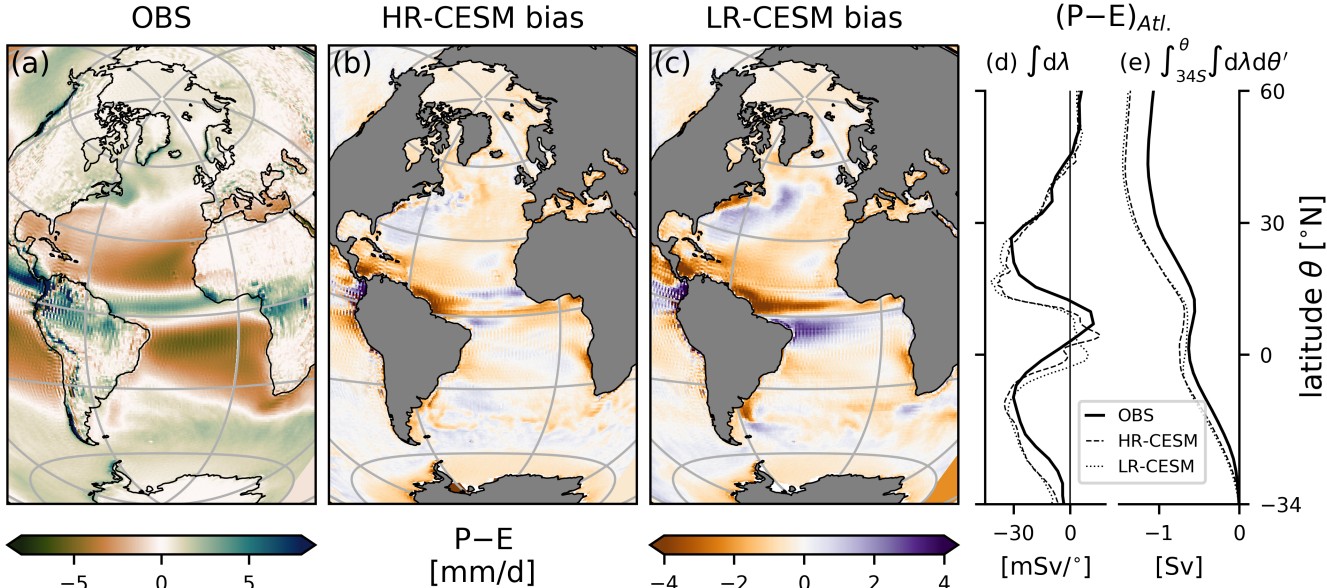

**Figure 2.** Precipitation minus evaporation: the observed ERA-Interim 1980-2010 climatology (a), and the biases of the HR-CESM (b) and LR-CESM (c) CTRL simulations. The zonally integrated $P - E$ fluxes per degree latitude (d) and the implied freshwater transports due to the $P - E$ fluxes (e) assuming zero transport at 34°S and constant salt content in the oceans. Note that this does not include runoff freshwater contributions.

et al., 2013). To compare to the model data, we first interpolated the EN4 data bilinearly horizontally and then linearly to the model depth coordinates for both HR- and LR-CESM ocean grids. Figure 3 shows the observed salinity distribution and the simulation biases of the upper 100 m, as an Atlantic zonal mean, and along a zonal transect at 34°S. Both HR-CESM (Fig. 3b) and LR-CESM (Fig. 3c) show a similar bias pattern with a positive surface bias in the North Atlantic and parts of the Arctic, and a negative bias in the South Atlantic and the rest of the global ocean (supplementary Fig. A3). The Atlantic zonally averaged profile (Fig. 3d) shows the largest salinity values in the evaporative subtropical gyres with less saline waters of the NADW with $S = 34.9 - 35.0$ at high northern latitudes and between 1500 and 4000 m while relatively fresh AAIW is visible around 1000 m depth up to 10°N. The AAIW is visible in the 34°S transect (Fig. 3g) and the section average (Fig. 3j). The bias in HR-CESM is significantly reduced compared the LR-CESM simulation in all three planes. In particular, the LR-CESM simulation bias around 1000 m depth and 15-30°N (Fig. 3f) can be attributed to the absence of eddies (Treguier et al., 2014).

### 2.2.4 Circulation and gateway transports

In Fig. 4a-c we compare the standard deviation of the observed sea surface height (SSH; Zlotnicki et al. (2019)) and the modeled dynamic sea level to illustrate the fidelity of the HR-CESM simulation. The SSH observations are provided as 5-daily means on a 1/6° grid and some polar data is missing. For Fig. 4a we use the year 2018 to calculate the standard deviation

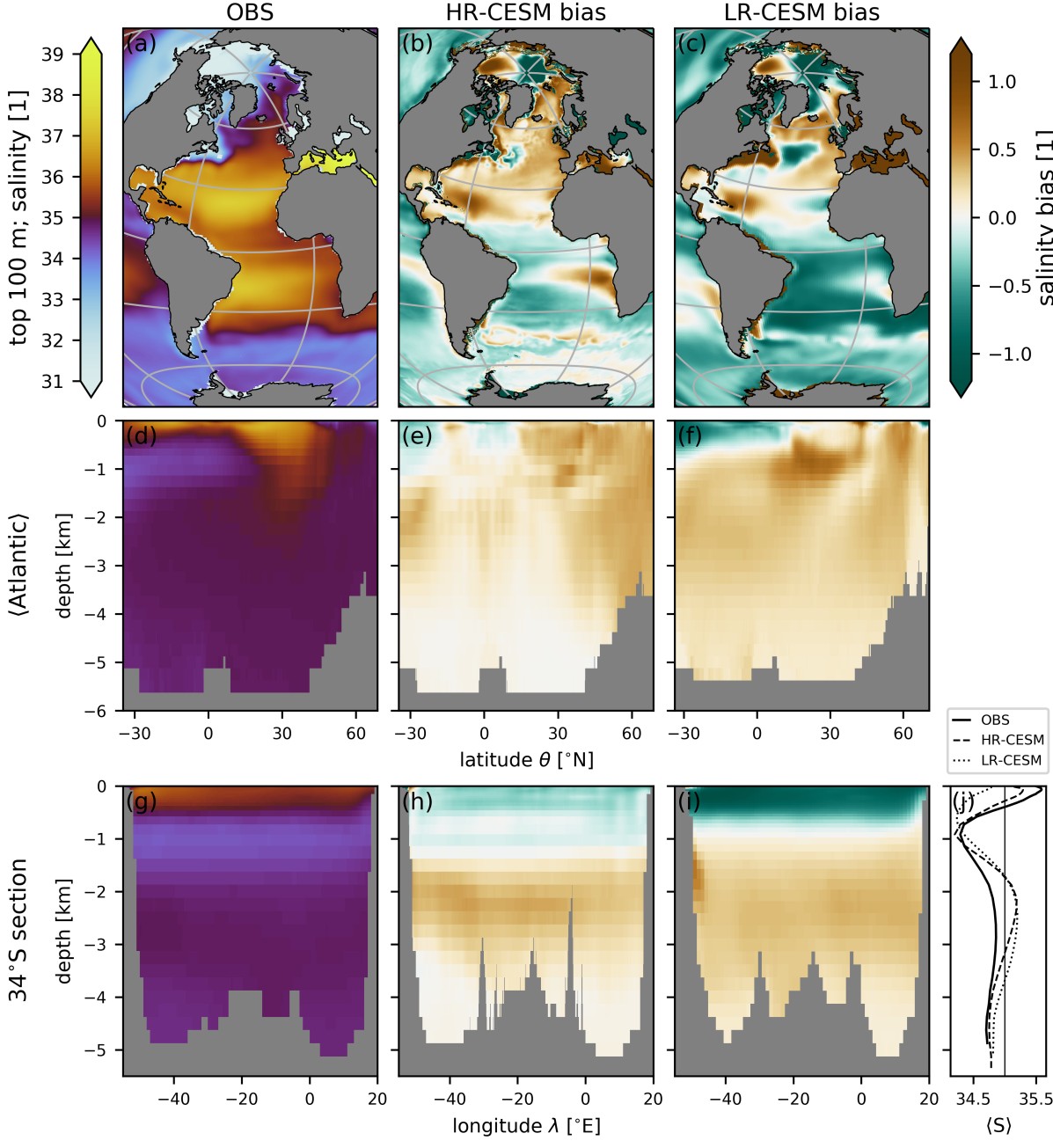

**Figure 3.** The salinity distribution in the EN4 observations (left) and the bias of the HR-CESM (center) and LR-CESM (right) CTRL simulations for the top 100 m (top), zonally averaged in the Atlantic (middle), and at the 34°S transect (bottom). Panel (j) shows zonally averaged salinity profiles for observations (thick solid line) and the HR-CESM (dashed) and LR-CESM (dotted) simulations together with the reference salinity $S_0 = 35$ (thin solid). Note that salinity as a mass fraction is dimensionless.

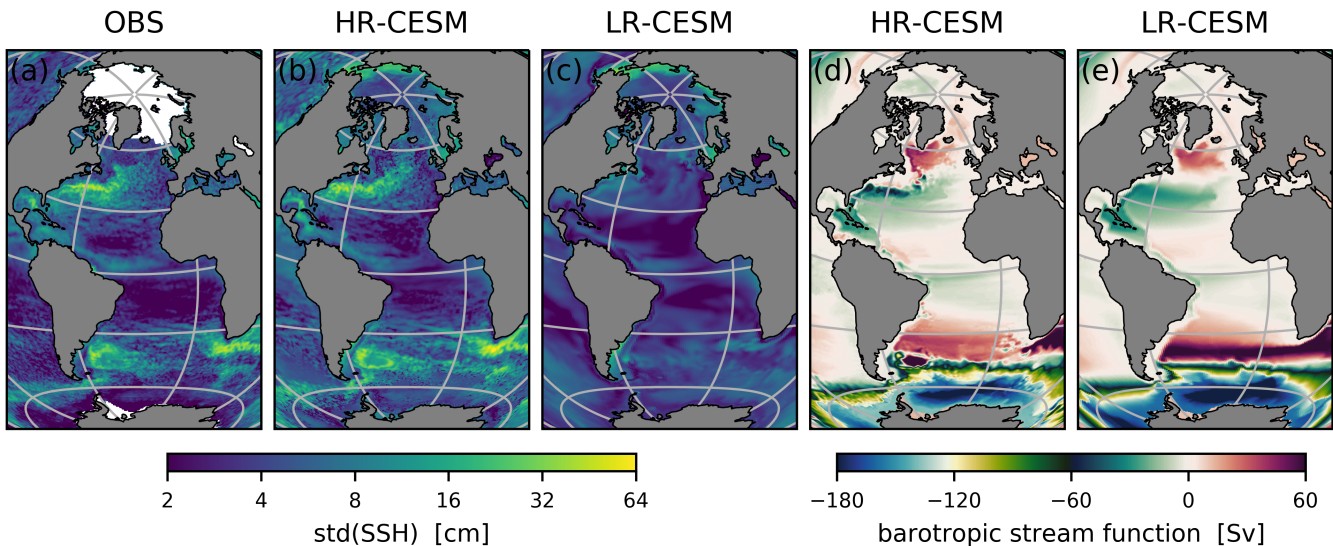

**Figure 4.** (a) The standard deviation of the observed sea surface height anomalies on a $1/6°$ grid (Zlotnicki et al., 2019). Missing data is white. (c/d) The dynamic sea level standard deviation of the HR- and LR-CESM CTRL simulations. (d/e) The mean barotropic streamfunction $\Psi$ of the HR- and LR-CESM CTRL simulations relative to the African Atlantic coast.

from the 5-daily means. In Fig. 4b/c we calculate the standard deviation for the branch-off year of the CTRL simulations based on the $\text{SSH}^2$ and $\text{SSH}$ output variables. The means that the modelled dynamic sea level standard deviation uses the model time step as the sampling frequency, in effect capturing more variability than the 5-daily sampling of the observations. The observations and model are thus not directly comparable in all details, but the LR-CESM clearly lacks variability compared to the observations and HR-CESM, in particular in regions of the Gulf Stream and its extension, the Agulhas retroflection, the Argentine Basin and the Antarctic Circumpolar Current.

To explain several of the HR- and LR-CESM salinity distribution biases, we plot the barotropic streamfunction for both CTRL cases in Fig. 4d/e. We approximate the barotropic streamfunction $\Psi$ as $\Psi(\lambda, \theta) = \int_{\theta'=\theta_S}^{\theta} \int_{z=D}^{0} u(\lambda, \theta', z) \, \mathrm{d}z \, \mathrm{d}\theta' - \Psi_0$. Here the vertical integral of the zonal velocity $u$ is taken over the full depth from $z = D$ to the surface $z = 0$, the meridional integral is taken from the southern boundary at Antarctica $\theta = \theta_S$, and is subsequently set to 0 at the African Atlantic coast by removing a constant $\Psi_0$. In LR-CESM, the barotropic streamfunction is diagnosed and part of the model output and this field agrees well with our approximation. For consistency we present our approximation for both HR- and LR-CESM.

While the broad scale wind-driven subtropical and subpolar gyre circulation are present in both simulations, HR-CESM features stronger boundary currents, standing eddies, a more realistic Agulhas retroflection pathway and Gulf Stream separation point, and a stronger subpolar gyre which extends much further south along the North American coast. In LR-CESM, the inflow of Indian Ocean waters is unrealistically strong and together with the strong upper $100\,\mathrm{m}$ fresh bias of the Indian Ocean (supplementary Fig. A3) contribute to the negative salinity bias of the South Atlantic (Fig. 3). In the RCP scenario the Gulf

Stream in HR-CESM shifts northward which is expected under climate change (Yang et al., 2020). In LR-CESM the subpolar gyre weakens broadly, while in HR-CESM only the boundary currents weaken (not shown).

The freshwater fraction $W$, which we call freshwater for brevity, is defined relative to a reference salinity $S_0$ as $W = (S - S_0)/S_0$ where $S$ is the (dimensionless) salinity of a given ocean water parcel. We choose $S_0 = 35$ as this is the salinity of the modeled North Atlantic Deep Water and close to the average salinity at 34°S (cf. Figs. 3d/g/j). In principle, the freshwater framework has disadvantages, as the choice of the reference salinity $S_0$ is arbitrary and the amount of freshwater depends non-linearly on it (Schauer and Losch, 2019). However, the relevant terms relate to recirculating and eddy flows which are independent of $S_0$ (see App. B) and the AMOC stability criterion $F_{ovS}$ is framed in terms of freshwater. Only the barotropic transport depends on $S_0$ and this component is negligible for the Atlantic freshwater transport although it contributes significantly to the total salt transport.

**Table 2.** Transports into the Atlantic of sea water, salt, and freshwater through Bering Strait (Woodgate and Aagaard, 2005), the Strait of Gibraltar (Sánchez-Román et al., 2009), and across 24°S (Bryden et al., 2011). The barotropic sea water volume transport $F_{bt}^V$ is given for all three transects and the overturning volume transport $F_{ov}^V$ is given for the Strait of Gibraltar and at 24°S. The mean salt transport $F_{mean}^S$ across a section is defined as the integrated product of monthly salinity and velocity fields (the unit $[\mathrm{kt\,s^{-1}}]$ is equivalent to the also commonly used $[\mathrm{Sv\,psu}]$). The mean freshwater transport $F_{mean}$ is defined similarly with the reference salinity $S_0 = 35$. The freshwater transport overturning component $F_{ov}$ is given only at 24°S. The $\pm$ denotes uncertainties in the observations, and inter-annual standard deviations in the simulations. The two values for the observations at 24°S are estimates from two separate cruises.

| transect | term | units | OBS | | HR-CESM | | LR-CESM | |
|---|---|---|---|---|---|---|---|---|
| Bering | $F_{bt}^V$ | [Sv] | 0.8 | ±0.1 | 1.6 | ±0.1 | 1.0 | ±0.1 |
| | $F_{mean}^S$ | $[\mathrm{kt\,s^{-1}}]$ | 26 | ±3 | 51 | ±4 | 32 | ±1 |
| | $F_{mean}$ | [mSv] | 140 | ±20 | 100 | ±30 | 100 | ±30 |
| Gibraltar | $F_{bt}^V$ | [mSv] | −38 | ±7 | 0 | ±4 | 0 | ±3 |
| | $F_{ov}^V$ | [mSv] | 800 | | 430 | ±31 | 300 | ±25 |
| | $F_{mean}^S$ | $[\mathrm{kt\,s^{-1}}]$ | 0 | | 1.1 | ±0.2 | 1.1 | ±0.1 |
| | $F_{mean}$ | [mSv] | −50 | | −32 | ±2 | −32 | ±2 |
| 24°S | $F_{bt}^V$ | [Sv] | −0.755 | −0.630 | −1.6 | ±0.1 | −1.0 | ±0.1 |
| | $F_{ov}^V$ | [Sv] | 21.5 | | 17.0 | ±0.8 | 16.4 | ±0.7 |
| | $F_{mean}^S$ | $[\mathrm{kt\,s^{-1}}]$ | +0.3 | | −67 | ±5 | −47 | ±3 |
| | $F_{mean}$ | [mSv] | −2.9 | −2.4 | 290 | ±13 | 340 | ±14 |
| | $F_{ov}$ | [mSv] | −130 | −90 | 160 | ±10 | 270 | ±10 |

For the Atlantic-Arctic basin the only oceanic exchanges of freshwater and salt north of 34°S occurs at Bering Strait with the Pacific Ocean and through the Strait of Gibraltar with the Mediterranean Sea. Table 2 summarizes the observed and simulated transports of sea water, salt, and freshwater. The Mediterranean is a net evaporative basin with a small net volume inflow at the Strait of Gibraltar but an overturning that is about 20× stronger, importing salinity 36.2 waters and exporting Mediterranean Overflow Water at salinity 38.4 at a depth of 1000 m (Sánchez-Román et al., 2009). In reality there is no source of salt in the

Mediterranean, but as the model formulation is volume conserving, there is no net flow through the Strait of Gibraltar and the net evaporation in the model Mediterranean represents a virtual salt source. An overturning of $0.8\,\mathrm{Sv}$ with the aforementioned salinity differences would result in a salt transport of $1.8\,\mathrm{kt\,s^{-1}}$ into the Atlantic. The model salt transports of both simulations

are somewhat smaller at $1.2 \pm 0.1\,\mathrm{kt\,s^{-1}}$ which is equivalent to a freshwater transport of $32 \pm 2\,\mathrm{mSv}$. Through the shallow Bering Strait relatively fresh water with mean salinity of $32.5 \pm 0.3$ flows northward into the Arctic Ocean because of dynamic sea level differences between the Arctic and North Pacific (Woodgate and Aagaard, 2005).

Table 2 also lists the transport terms at 24°S as Bryden et al. (2011) provide $F_{ov}$ estimates from two cruises at this latitude. The simulated barotropic volume transport at 24°S equals that through Bering Strait, because the model is volume conserving.

The corresponding virtual salt flux formulation is also the reason why the simulated mean salt transport is very negative while observations show a small northward salt transport. Both simulations' overturning circulation is weaker than observed and the freshwater transport due to the overturning is of opposite sign compared to the observations.

## 3 Results

### 3.1 AMOC

Figure 5a/b shows the AMOC streamfunction $\psi(\theta, z)$ of the CTRL mean state (shading) together with the RCP trends (contours) in both HR- and LR-CESM. The maximum of $\psi$ is located just below $1000\,\mathrm{m}$ depth for both simulations around 35°N, but the LR-CESM simulation upper cell stretches further north consistent with its STG that extends too far north (cf. Fig. 4). The Antarctic Bottom Water cell is stronger and extends further north in HR-CESM. Both simulations experience a similar weakening and shoaling trend of the upper cell and a slight strengthening of the lower cell. The HR-CESM latitudinal gradient

in the weakening trend around the maxima at $2000\,\mathrm{m}$ is weaker so that the HR-CESM AMOC weakening is stronger at 34°S but weaker in the Northern Hemisphere compared to the LR-CESM simulation.

The evolution of the AMOC strength is measured at the latitude of the RAPID mooring array, 26.5°N, and the depth of maximum overturning, $1000\,\mathrm{m}$ (white crosses in Fig. 5a/b). In contrast to the RAPID array data, the streamfunction in Fig. 5c contains negligible contributions from the Gulf of Mexico at 26.5°N. The panels 5d/e show time series of the AMOC strength,

including 100 years prior to the RCP branch-off point to show the the statistically equilibrated nature of the time series. Both simulations' CTRL mean AMOC strength compare favorably to the observations with approximately $18\,\mathrm{Sv}$ (Frajka-Williams et al., 2019) and they respond with a similar linear weakening trend of $4.7\,\mathrm{Sv/century}$ and $5.2\,\mathrm{Sv/century}$ to the RCP forcing, respectively. The monthly variability (thin line) of HR-CESM is larger than in LR-CESM, due to the presence of an eddying ocean.

### 3.2 Surface freshwater fluxes

The Atlantic is a net evaporative basin and Fig. 6 shows maps of the total surface freshwater flux, $F_{surf}$, and its major contributing components: precipitation $P$ and evaporation $E$. In addition, $F_{surf}$ comprises runoff from land $R$ and ice, as

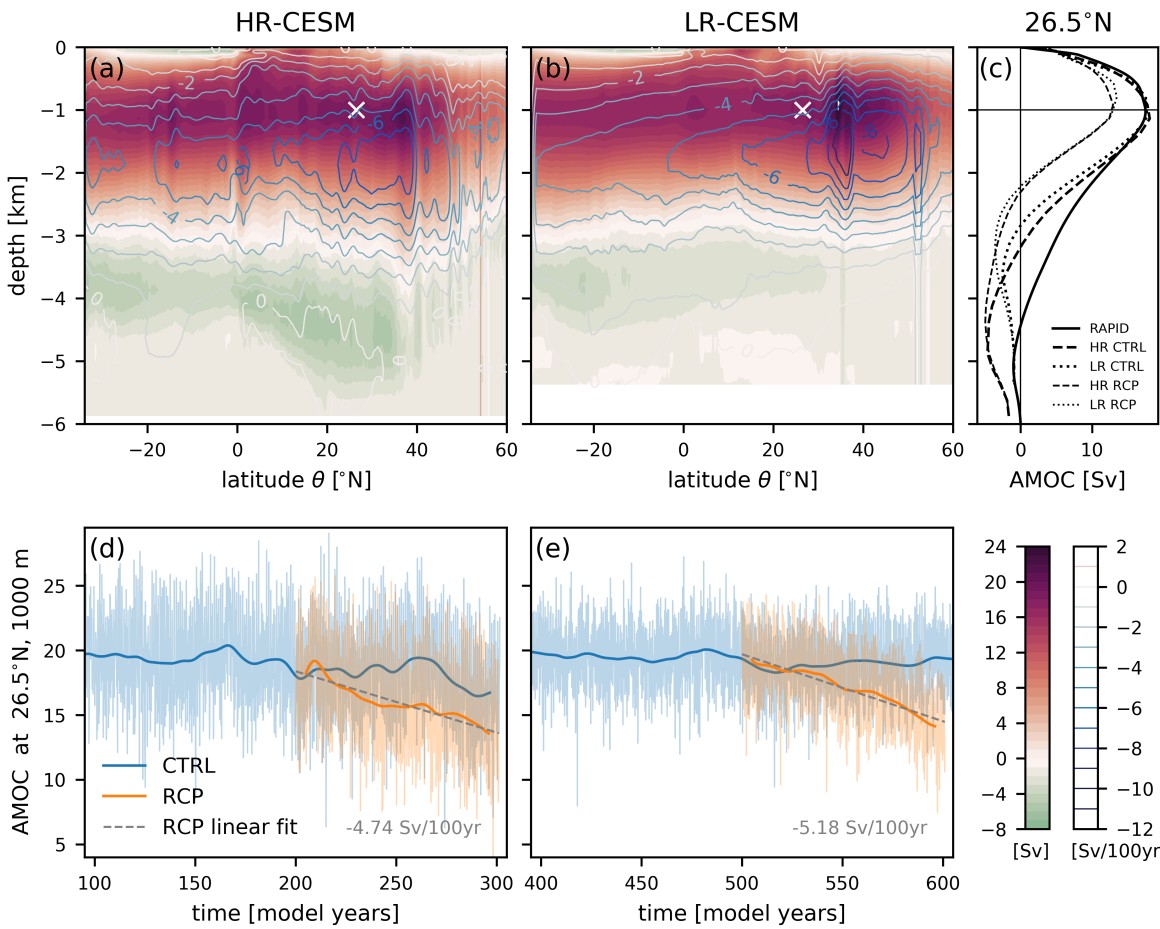

**Figure 5.** AMOC mean streamfunctions $\psi(\theta, z)$ of the HR-CESM (a) and LR-CESM (b) CTRL simulations together with the linear trends as contour lines every $1\,\mathrm{Sv/century}$. At 26.5°N and $1000\,\mathrm{m}$ depth, the white crosses mark the location of AMOC strength whose time evolution is depicted in (d) and (e). Panel (c) compares the modeled CTRL mean streamfunction profiles at 26.5°N (thick dashed/dotted) to the RAPID observations (solid) and also presents the changed streamfunction after 100 years in the RCP simulation (thin dashed/dotted). Monthly (thin) and 10 year lowpass filtered (thick) AMOC time series of the HR-CESM (d) and LR-CESM (e) CTRL (blue) and RCP (orange) simulations. The linear trends are indicated by grey dashed lines and their values are written in the lower right.

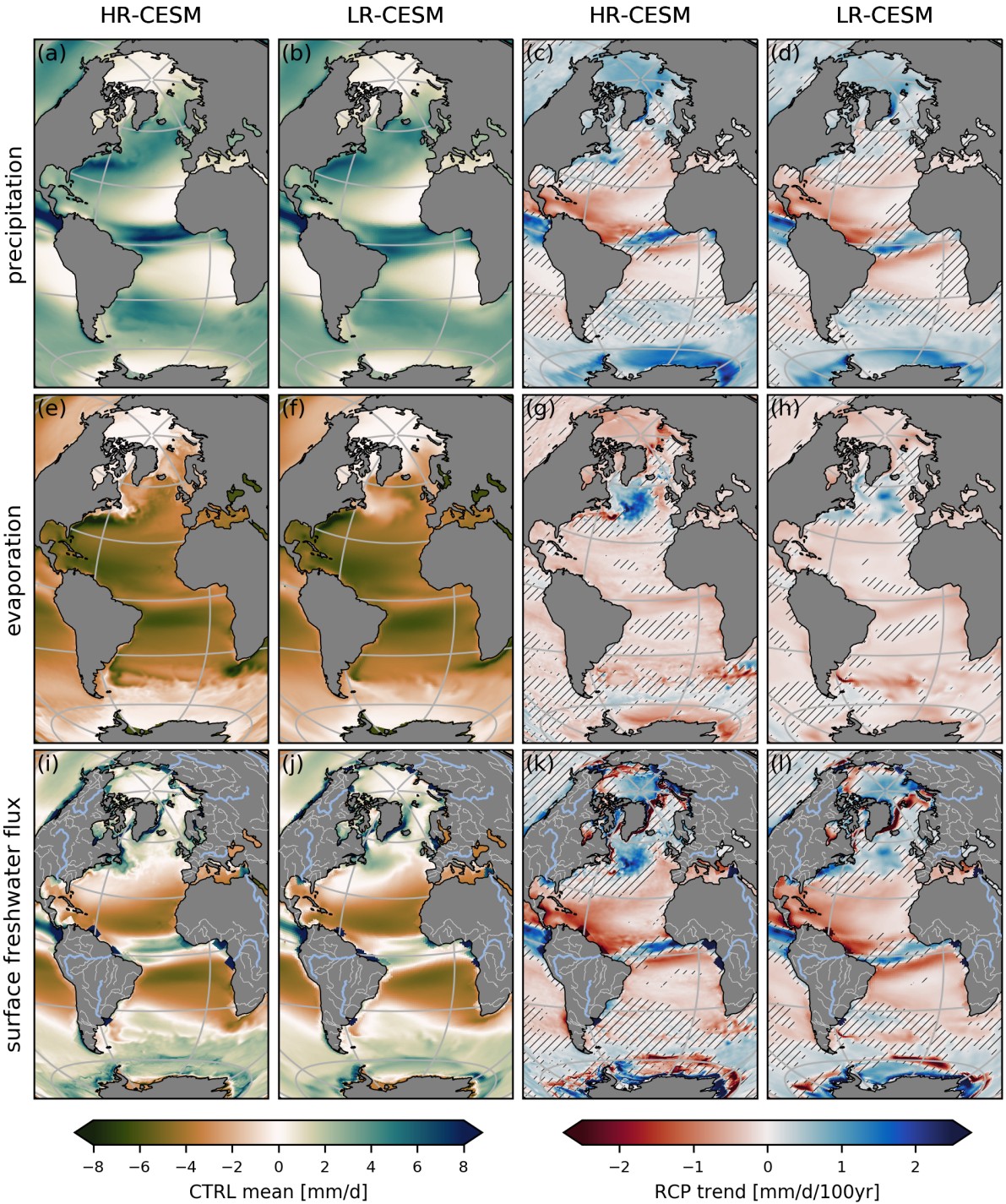

**Figure 6.** The major freshwater flux components, precipitation $P$ (tow row) and evaporation $E$ (middle), and the total freshwater flux $F_{surf}$ (bottom). The means of the HR-CESM (left column) and LR-CESM (center left) CTRL simulations and the linear trends of the HR-CESM (center right) and LR-CESM (right) RCP simulations. Polygons near river mouths in panels (i-l) are areas where runoff is distributed by the ocean model. Hatched areas in the trend maps are not significant at the 5% level.

well as sea ice melt (brine rejection) which, from here on, are all defined as positive (negative) freshwater fluxes into the ocean. Precipitation occurs mainly in the ITCZ region, with stronger maxima in HR-CESM over the Gulf Stream, and in the midlatitude storm tracks. In the HR-CESM (LR-CESM) CTRL simulation between 34°S and 60°N there is a net freshwater loss of $0.85\,(0.93)\,\mathrm{Sv}$ which is relatively large compared to the CMIP5 freshwater loss (e.g. $0.48\pm0.13\,\mathrm{Sv}$ in the historical multi-model mean in Fig. 6 of Skliris et al. (2020)). Evaporation is strongly tied to SSTs (cf. Figs. 1a-c and 6e/f) with most occurring in the subtropical gyres and at the above zonal-average SSTs of the western boundary currents and their extensions. Runoff occurs from all coasts bordering the Atlantic and is artificially distributed over larger areas surrounding the river mouths in the model which is visible in the total freshwater flux subplots of Fig. 6.

The forced freshwater flux trends in Fig. 6 reveal the general intensification of the hydrological cycle as SSTs generally increase (Fig. 1d/e). Total surface freshwater flux linear trends of $-0.14\,(-0.16)\,\mathrm{Sv/century}$ between 34°S and 60°N (cf. Fig. 9a) in the HR-CESM (LR-CESM) RCP simulation intensify the Atlantic's evaporative nature. The notable exception to this global trend in both simulations is the subpolar gyre where SSTs decline which results in less evaporation and hence a larger net freshwater flux into the ocean. This reduction in evaporation in the SPG is more pronounced in HR-CESM compared to LR-CESM. Regional differences between the simulations include smaller positive trends along the US East coast due to the different Gulf Stream separation behavior and the related southward extent of the subpolar gyre. The runoff into the Atlantic and Arctic increases almost everywhere with the exception of Amazon basin rivers. In the polar regions, changing freshwater input from melting sea ice is locally significant, e.g. east of Greenland where the total freshwater trends (Fig. 6c/d) are more negative than the evaporative component alone would suggest (g/h) despite increases in precipitation (c/d) due to higher atmospheric temperatures. The trends of the surface freshwater flux components are similar on a large scale between HR-CESM and LR-CESM (see Fig. 9).

### 3.3 Meridional transport of freshwater

We decompose the meridional freshwater transport into different terms related to the overturning and azonal gyre circulation as well as an eddy component (equations in Appendix B). Budget term computations are performed on the original ocean model grid which leads to small differences between model zonal transects and the true parallel of a given latitude in the mid- to high-latitudes of the Northern Hemisphere (green lines in Fig. 1b/c). Figure 7 shows the meridional dependence of the different zonally integrated northward freshwater transport components. The figure includes both the 30 year CTRL means and the year 2100 values of the linear RCP trends (top row), as well as the trends themselves (bottom row).

At 60°N, the total freshwater transport $F_{tot}$ (red lines in Fig. 7) is negative because relatively fresh water is imported via Bering Strait into the Arctic where it further freshens mostly due to runoff (Tbl. 2). Despite different volume fluxes at Bering Strait, the freshwater inflow is about the same between the simulations at $0.10\pm0.03\,\mathrm{Sv}$ because of the stronger fresh bias of the LR-CESM simulation (cf. Tbl. 2, Bering Strait salinity bias in Fig. 3b/c, and vertical lines in Fig. 7). The Arctic is a net precipitative basin, in part due to its extensive catchment area, resulting in even more freshwater entering the Atlantic at 60°N. In the subpolar gyre and the ITCZ, i.e. in latitudes of net precipitation, freshwater diverges (i.e. $\partial F_{total}/\partial\theta > 0$), while net evaporation in the subtropical gyres results in freshwater convergence by the oceanic transport. Under the RCP forcing scenario,

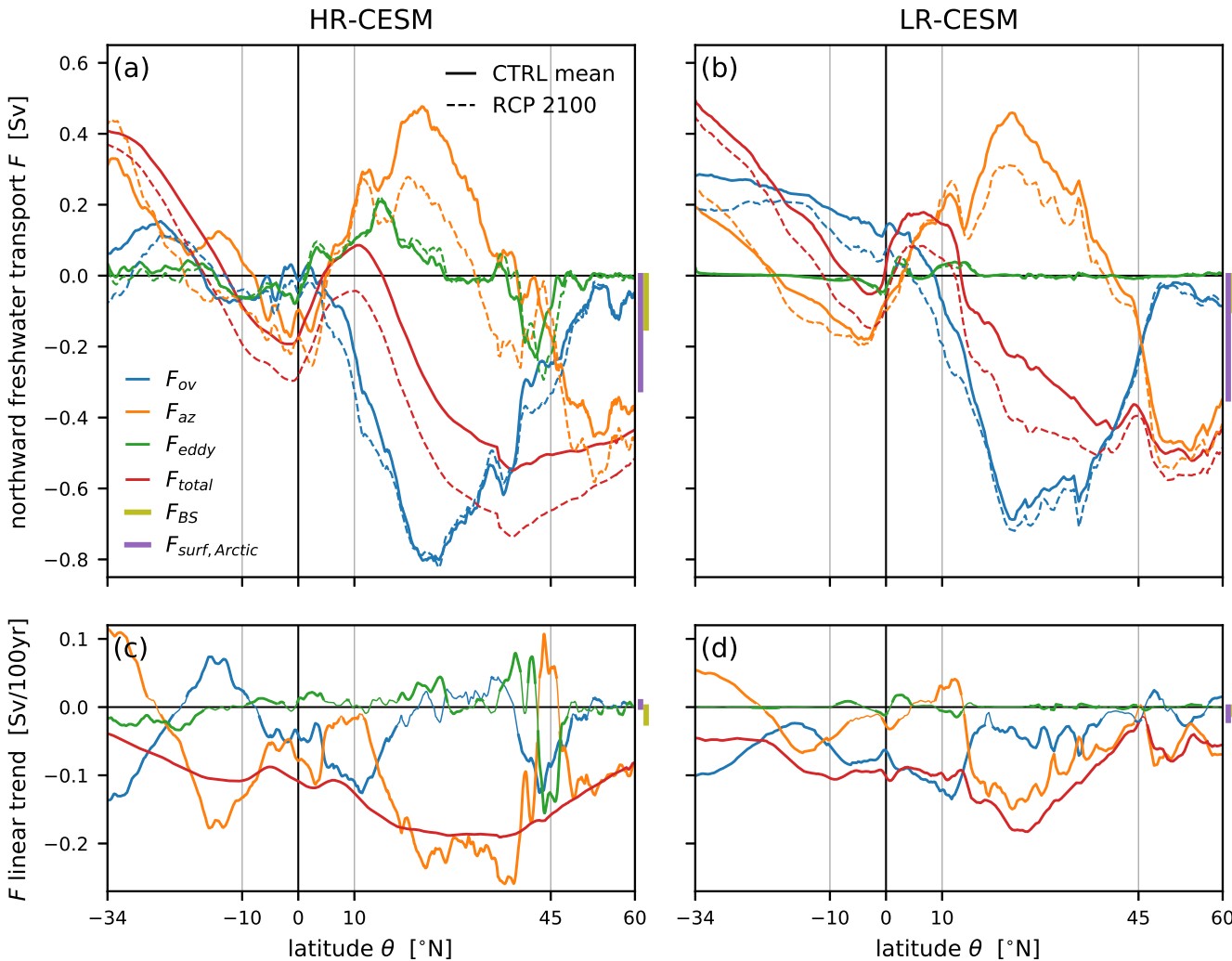

**Figure 7.** The meridional freshwater transports as a function of latitude $\theta$ for the HR-CESM (left) and LR-CESM (right) simulations. In (a) and (b), solid lines are the means of the 30 CTRL years following the branch-off point and dashed lines are the year 2100 values of the linear RCP fit. The total transport terms (red) are decomposed into an overturning (blue), an azonal gyre (orange), and an eddy contribution (green). Panels (c) and (d) show the linear trends separately. Significant trends (at the 5% level) are thick, while insignificant trends are only thin. Vertical bars to the right of the panels illustrate the Arctic surface flux in the Arctic (purple) and the Bering Strait inflow (olive) of freshwater.

the total freshwater flow is more southward because more freshwater enters at 60°N primarily due to increased net precipitation (including runoff) in the Arctic. Generally, meridional gradients of the total transports in precipitative and evaporative latitudes increase as a result of the enhanced hydrological cycle. Notable differences between the HR- and LR-CESM $F_{tot}$ are at the
STG-SPG boundary at 45°N where the LR-CESM $F_{tot}$ does not exhibit the negative transport trend of the HR-CESM $F_{tot}$ and the meridional position of the tropical freshwater transport divergence related to the southward biased ITCZ position of LR-CESM. In the following we take a closer look at these two differences.

The AMOC carries both relatively salty surface waters and fresh Antarctic Intermediate Waters northward and salty North Atlantic Deep Water south. The overturning freshwater transport $F_{ov}$ (blue lines in Fig. 7) thus depends on the vertical dis-
tribution of the zonally averaged salinities relative to the depth of the overturning cell (cf. Figs. 3 and 5). Without changes to salinity, a weakening AMOC would reduce the overturning transport, and with a constant AMOC, the intensifying hydrological cycle would lead to enhanced meridional gradients in the transport across precipitative and evaporative parts of the ocean. With the weakening AMOC under the RCP scenario, the $F_{ov}$ trend is negative everywhere in LR-CESM, while the HR-CESM $F_{ov}$ trend is not latitudinally coherent in its sign. The HR-CESM $F_{ov}$ decrease around 40-50°N is caused by the northward
migration of the boundary between the subtropical and subpolar gyres. The decrease in the overturning transports has some of its largest expression at 34°S and the salinity stratification bias of the South Atlantic (Fig. 3j) results in a positive $F_{ovS}$ bias which is more pronounced in LR-CESM compared to HR-CESM.

In the absence of eddies, the decomposition of the total flow into the overturning and azonal component depends on the azonality of the velocity and salinity fields. Both the North and South Atlantic subtropical gyres transport freshwater north
(orange lines in Fig. 7) due to their opposite zonal asymmetry in salt content near the surface where the majority of horizontal gyre transport takes place (Fig. 3) while the subpolar gyre transports freshwater south. Boundary currents, which comprise an important part of the azonal flow, are better resolved in the HR-CESM simulation (cf. Figs. 1-4). Under the climate change scenario, the azonal freshwater transport term $F_{az}$ generally becomes more southward north of 20°S in the STGs and the SPG. The gyre transport trends consist both of a gyre strength signal (approximately the barotropic stream function of Fig. 4) and
one due to the salinity azonality trend (Fig. 8). The HR-CESM $F_{az}$ trends are the largest contribution to the total southward freshwater trends. In fact, between 20-40°N the HR-CESM $F_{az}$ trend is so negative due to the strong salinification along the North American Atlantic coast (Fig. 8), that the $F_{ov}$ trend becomes slightly positive (but not significantly so). This occurs also around 5-20°S in the HR-CESM simulation where $F_{az}$ switch signs under forcing. These negative $F_{az}$ trends are much weaker in LR-CESM so that the overturning component trend remains latitudinally coherent in its sign.
Eddy transports of freshwater $F_{eddy}$ (green lines in Fig. 7) are not associated with volume fluxes as they are due to correlations between salinity and flow anomalies, which we define with a cutoff time scale of one year, i.e. including the seasonal cycle. The supplementary Figure B1 shows the effect of using a monthly cutoff time scale. A detailed analysis of the eddy salt transports in the Atlantic revealed that they are associated with two distinct mechanisms (Treguier et al., 2012). First, at the STG equatorward edges seasonal variations in surface salinity and wind driven circulation cause eddy transports. Second, at
the boundary between the subtropical and subpolar gyres baroclinic mesoscale eddies are responsible for eddy transports. As expected, in the diffusive LR-CESM the eddy transports are negligible outside tropical seasonal variability, but in HR-CESM,

the eddy freshwater transports $F_{eddy}$ contribute significantly and bring freshwater polewards in the low latitudes and equatorwards around the Gulf Stream and its extension. The eddy transports thus move freshwater generally down-gradient, which is parameterized in LR-CESM with the Gent-McWilliams scheme as a diffusive salt flux (Gent and McWilliams, 1990). Under 340 the RCP scenario, there is essentially no change in the small LR-CESM eddy transports, but the HR-CESM eddy transport magnitude changes markedly around 45°N where the Gulf Stream shifts northward (Fig. 4) and the meridional salinity gradient increases (Fig. 8). In contrast to HR-CESM, freshwater diverges around 40-45°N in LR-CESM due to the absence of eddy transports. This LR-CESM freshwater divergence contributes to the salinity bias (cf. Fig. 3f).

## 3.4 Salinity trends

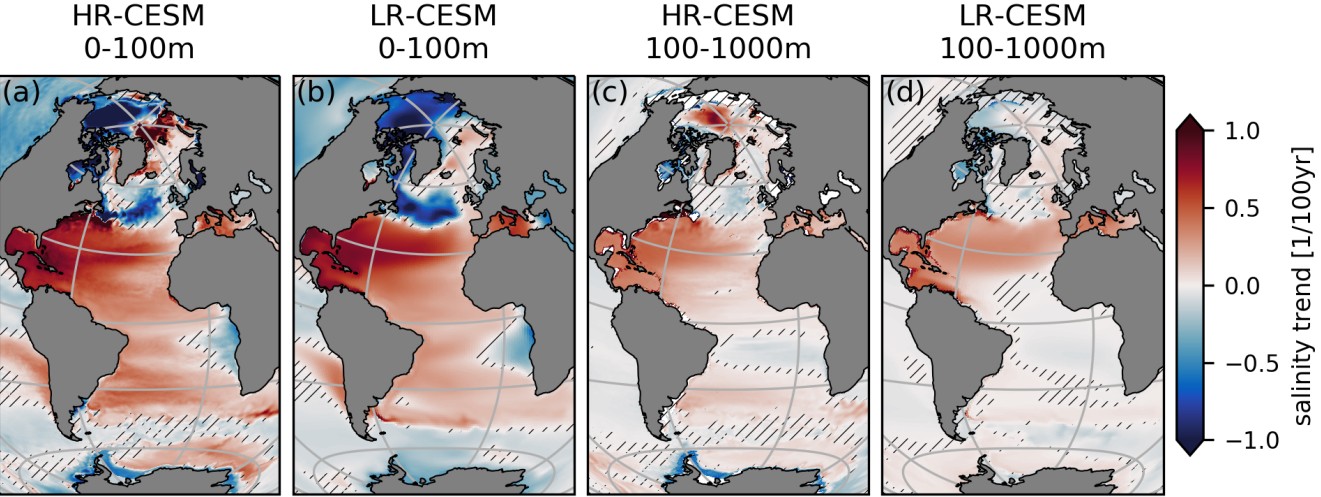

**Figure 8.** Linear trends of the vertically averaged salinity in the surface $(0 - 100\,\mathrm{m})$ and subsurface $(100 - 1000\,\mathrm{m})$ layers under the RCP scenario. Areas where the linear trend is not significant at the 5% level are hatched.

In the RCP climate change scenario, the Atlantic's salinity changes significantly as surface freshwater fluxes and transport convergences change, even though these salt storage changes are small compared to the fluxes and their changes. Figure 8 shows the linear trend of the vertically averaged salt content for the surface $(0 - 100\,\mathrm{m})$ and subsurface $(100 - 1000\,\mathrm{m})$ layers. In the forced salinity response the signature of the enhanced hydrological cycle is imprinted: the upper $1000\,\mathrm{m}$ of the Atlantic south of 45°N largely salinifies, in particular in the NA-STG. The freshening of the NA-SPG is also a consequence of the 350 weakening AMOC and the associated warming hole (Menary and Wood, 2018). The surface subpolar gyre freshens uniformly in the LR-CESM simulation, but the subsurface shows largely insignificant trends. In HR-CESM, only the eastern SPG freshens down to $1000\,\mathrm{m}$ but salinifies in the East and West Greenland as well as Labrador Currents bringing salt into the western SPG. This is the result of advection of salinifying waters from the central Arctic north of Greenland and Svalbard. While the Arctic surface layer between Bering Strait and the North Pole becomes fresher in both simulations due to enhanced net precipitation

including runoff (Fig. 6), the subsurface salinifies strongly in HR-CESM enhancing stratification. In the Southern Hemisphere, enhanced runoff from Africa decreases the salinity in the eastern SA-STG whereas decreasing runoff from South America enhances the salinification downstream of the Brazil and North Brazil Currents (see Fig. 6).

The zonal gradient of the salinity trends of the upper $1000\,\mathrm{m}$ in Fig. 8 is generally westward equatorward of $45°$ and more pronounced in HR-CESM. This leads to more azonal northward salt and southward freshwater transport by the North Atlantic subtropical gyre and where the southward Angola Current carries enhanced runoff from tropical Africa southward (Fig. 6k/l). South of $25°$S the trend enhances the existing zonal salinity gradient resulting in strengthened azonal transport components (Fig. 7). The azonality at $34°$S is opposed by surface freshwater flux trends at this latitude. This is more strongly so in LR-CESM compared to HR-CESM (Fig. 6) where it leads to a weaker enhancement of the azonal transport components.

### 3.5 Freshwater budget

In order to gain insight into regional changes, we evaluate the freshwater budget over several regions of the Atlantic and Arctic, which is formulated as

$$\frac{\mathrm{d}\bar{W}}{\mathrm{d}t} = F_\nabla + F_{surf} + F_{mix} \tag{1}$$

where the change in freshwater storage over time $\mathrm{d}\bar{W}/\mathrm{d}t$ over a region is a consequence of the freshwater convergence across the lateral volume boundaries $F_\nabla$, surface fluxes $F_{surf}$, and a residual mixing term $F_{mix}$ that captures subgridscale diffusion (including eddy parametrizations) and errors introduced by our choice of the reference salinity $S_0 = 35$. The freshwater content $W$ of a volume $V$ of ocean water is defined relative to the reference salinity as $\bar{W} = -1/S_0 \int (S - S_0)\,\mathrm{d}V$. Similarly, freshwater transport across a surface is defined as $F = -\int \frac{S-S_0}{S_0}\boldsymbol{u}_\perp\,\mathrm{d}A$ where $\boldsymbol{u}_\perp$ is the velocity perpendicular to the surface element $\mathrm{d}A$. Surface freshwater fluxes, $F_{surf}$, are implemented as virtual salt fluxes, $F^S_{surf}$, in the POP2 model and we calculate this flux as $F_{surf} = -F^S_{surf}/S_0$.

Figure 9 presents the freshwater budget terms for each of the regions and the whole Atlantic from $34°$S to $60°$N (boundaries as green lines in Fig. 1b/c). Panel (a) is a summary of the tendency and main freshwater flux terms into the ocean as in Eq. 1, panel (b) presents the constituent components in more detail with trends indicated, and panel (c) focusses on the trends of panel (b). The mean values of the CTRL simulations are represented by bars and the 100 year linear trends by arrows. The summary plots (a) show how the subtropical gyres are net evaporative and ocean currents converge freshwater there (negative purple $F_{surf}$ and positive red $F_\nabla$). On the other hand, both the ITCZ and the NA-SPG gain freshwater through surface fluxes. Here the freshwater transport divergence (red) is much smaller in magnitude compared to the STG freshwater convergences both due the smaller areas and flux densities (cf. Figs. 1 and 6i/j) and the STGs dominate the signal of the whole Atlantic from $34°$S to $60°$N. The freshwater reservoir tendency term $\mathrm{d}\bar{W}/\mathrm{d}t$ (cyan) is small compared to the other terms. However, for example for the whole Atlantic between $34°$S and $60°$N, the tendency term is crucial in closing the budget as the trends of the transport convergence, $-\nabla F_{tot}$, are smaller than the opposing trends in the surface fluxes. Full depth regionally integrated salt content trends are very similar between the simulations with largest salt content increase in the NA-STG (cf. Figs. 9a and 8). The mixing term $F_{mix}$ (brown) is negligible in HR-CESM but sizable in LR-CESM where it includes the parameterized

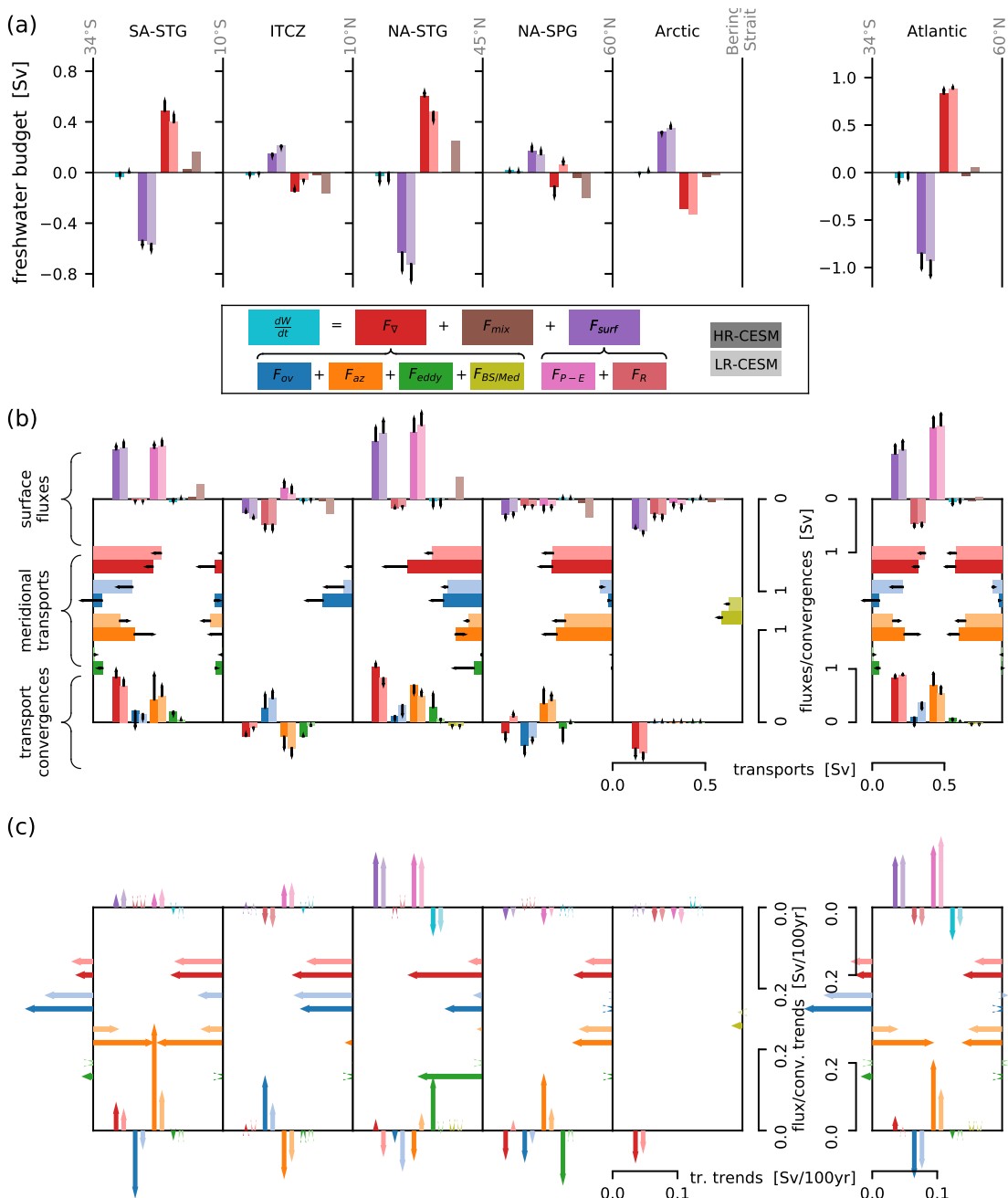

**Figure 9.** Integrated freshwater budget (Eq. 1) for different zonal bands of the Atlantic. The boundary latitudes (grey) are shown in Fig. 1. Bars are the CTRL simulation averages and arrows indicate the linear change in year 2100 of the RCP simulations. Panel (a) summarizes the freshwater budget for the regions with the terms of Eq. (1) with darker (lighter) colors representing HR-CESM (LR-CESM). The notation is explained in Appendix B and $\Delta \bar{W}$ is the change in freshwater content, over 30 years of the CTRL simulations and 100 years of the RCP simulations. Panel (b) shows the freshwater budget terms in detail where the horizontal bars are advective transports across the meridional boundaries with their convergence indicated by the vertical bars at the bottom. In addition the inflow from Bering Strait and the Mediterranean are shown. The bars at the top represent surface fluxes, the freshwater content change over time, as well as the mixing term and all bars are oriented such that inward pointing bars indicate addition of freshwater. Note that the vertical scale of the top and bottom bars is identical (if reversed), but the horizontal scale is different; as are the scales for the individual regions and the whole Atlantic.

diffusion by eddy fluxes which act down-gradient, thus adding freshwater to the saltier, evaporative STGs. The barotropic and hence total salt transport is southward everywhere due to the import through Bering Strait which is larger in HR-CESM, while the barotropic freshwater transport term sign and magnitude depends on the choice of $S_0$ (Schauer and Losch, 2019).

The top vertical bars of Fig. 9b show the major surface freshwater flux terms whereas the total (purple) is also presented in the summary plot above (Fig. 9a). As discussed with the surface flux maps (Fig. 6), both the means of the CTRL simulations (bars) and the RCP trends (attached arrows) are similar between the simulations given that the exact numbers depend on the choice of bounding latitude. South of 45°N, all chosen regions experience more evaporation than precipitation (Fig. 6), but in the ITCZ there is net freshwater flux into the ocean due to a large runoff especially from the Amazon and Congo rivers. The strongest trends exists in the NA-STG, but marked differences between the simulations' freshwater input trends exist only in the mid and high latitudes. In the SPG, the total freshwater input (purple) increases by approximately 20% in both the HR- and LR-CESM, but the HR-CESM experiences a stronger reduction in evaporation because of the lower SST trends (Fig. 1) which is offset by stronger runoff. In the Arctic, the HR-CESM freshwater input decreases slightly by 2% while the LR-CESM input increases by 5%. These relatively small numbers conceal a much larger enhancement of the HR-CESM hydrological cycle with precipitation (evaporation) increasing 55% (45%), while the LR-CESM precipitation (evaporation) only increase by 37% (24%).

The horizontal bars of Fig. 9b show the meridional transport components and bottom vertical bars their convergences. The total convergence (red) is also shown in the summary plot above (Fig. 9a). In steady state, the tendency term (cyan) $\mathrm{d}\bar{W}/\mathrm{d}t = 0$ and the total oceanic freshwater convergence (red) compensates the sum of the surface fluxes (purple). The magnitude of the regional convergences (red) is generally smaller for LR-CESM compared to HR-CESM. The HR- and LR-CESM differences of the overturning (blue) vs. azonal (orange) convergence decomposition offset each other in the STGs, the ITCZ, and the Atlantic as a whole, resulting in the same sign of the total transport convergence (red). Only in the SPG does the sign of the total convergence differ as the overturning convergence is stronger and the azonal divergence is weaker in HR-CESM and the mixing term captures the parametrized eddy transports in LR-CESM.

Figure 9c focusses on the trends of the transport terms while using the same layout as panel (b). Under the RCP scenario, the extreme strengthening of the HR-CESM eddy transport (green) at 45°N is related to the northward shift of the Gulf Stream under forcing (cf. Fig. 7; Yang et al. (2020)). The supplementary Fig. B1 shows that this negative eddy trend at 45°N consists to a large degree of a seasonal signal. The trends in the overturning and azonal convergence trends offset each other (except in the HR-CESM NA-STG with its strong growth in eddy convergence) indicating a change in the azonality of the salinity distribution (cf. Fig. 8). The barotropic and hence total salt transport is southward everywhere due to the import through Bering Strait which is larger in HR-CESM, while the barotropic freshwater transport term sign and magnitude depends on the choice of $S_0$ (Schauer and Losch, 2019).

## 3.6 AMOC stability indicators

The freshwater import (export) by the AMOC constitutes a negative (positive) feedback. The freshwater convergence by the overturning circulation $\Sigma = F_{ovS} - F_{ovN}$, where $F_{ovS}/F_{ovN}$ are located at 34°S/60°N, has been suggested as an indicator for

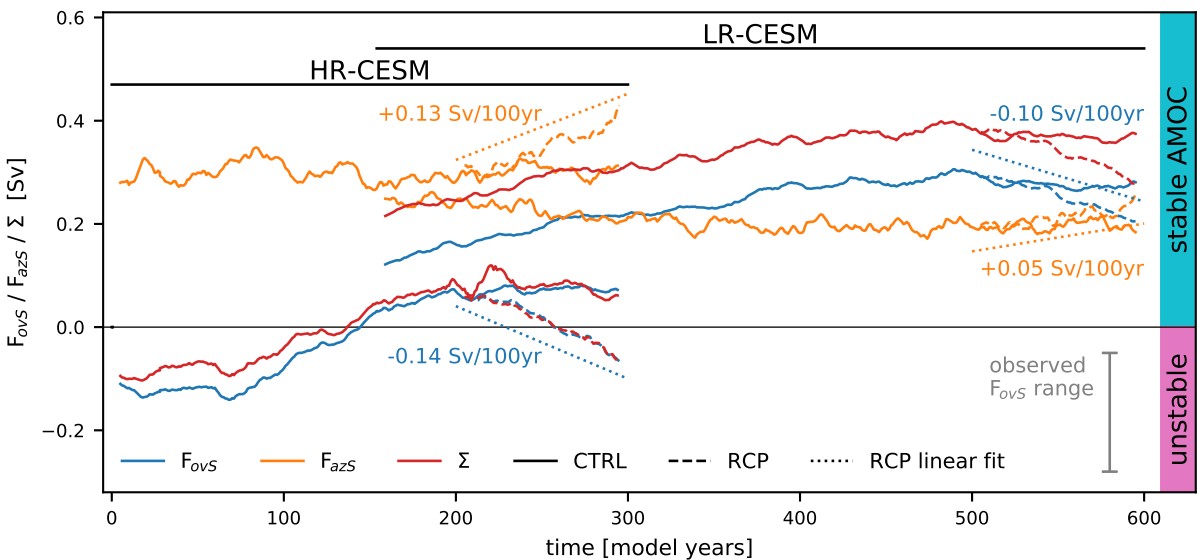

**Figure 10.** Time series of the annually averaged freshwater import into the Atlantic by the overturning circulation $F_{ovS}$ at 34°S (blue), the azonal freshwater transport contribution $F_{az}$ (orange), and the overturning freshwater transport divergence between 34°S and 60°N, $\Sigma$ (red). All available data is shown of the CTRL simulations (solid) together with the RCP simulations (dashed) branching off in year 200 (500) for HR-CESM (LR-CESM). The linear trend values of the RCP simulations are written close to offset linear fits (dotted). The observed $F_{ovS}$ range (grey) is from Weijer et al. (2019) and the AMOC stability regimes are labelled on the right.

an AMOC multiple equilibrium regime (Dijkstra, 2007; Huisman et al., 2010; Liu et al., 2014). Figure 10 shows the evolution of these indicators together with the azonal freshwater transport at 34°S, $F_{azS}$, for both CTRL and RCP simulations. Both the HR- and LR-CESM CTRL simulations initially equilibrate with increasing $F_{ovS}$ values (blue). At the point where the RCP

simulations are branched off, $F_{ovS}$ appears to have reached an equilibrium as the concurrent CTRL time series are statistically stationary. Despite a very similar overturning strength (Fig. 5), the LR-CESM CTRL $F_{ovS}$ values are significantly higher due to the stronger vertical salt bias (Fig. 3). Non-eddying CMIP5 models have a positive bias in the $F_{ovS}$ sign and may hence be too stable; much of this bias is a result of the salinity bias with fresh surface anomalies south of 20°N and salty anomalies elsewhere in the Atlantic (Mecking et al., 2017). Artificially replacing the CMIP5 model salinities by observed values as in

Mecking et al. (2017), reduces $F_{ovS}$ to negative values. The CTRL azonal component $F_{azS}$ (orange) equilibrates faster than the overturning component as it relates to the shallower transport by the wind-driven STGs. The total freshwater transport at 60°N is almost identical between the simulations and consists predominantly of the azonal component (cf. Figs. 7, 9), but the exact azonal vs. overturning decomposition differs such that the LR-CESM $F_{ovN}$ magnitude is larger than the HR-CESM $F_{ovN}$ magnitude, resulting in a larger offset in $\Sigma$.

In response to the RCP forcing, both HR- and LR-CESM exhibit negative $F_{ovS}$ trends at $-0.14$ and $-0.10\,\mathrm{Sv/century}$, respectively. The $F_{ovS}$ values decrease because the salinity trends offset the fresh bias near the surface (cf. Figs. 3 and 8). The $\Sigma$ value is also plotted in Fig. 10 and its trend is evidently dominated by $F_{ovS}$ while $F_{ovN}$ barely changes under forcing

(Fig. 7). The azonal gyre component $F_{azS}$ also evolves in response to the forcing (cf. Fig. 8) and is connected to $F_{ovS}$ through the overall freshwater budget (Cimatoribus et al., 2012). Its change compensates the change in $F_{ovS}$ completely in HR-CESM and only half of it in LR-CESM. Both $F_{ovS}$ and $\Sigma$ indicate a shift into the unstable, multiple equilibrium regime under the RCP forcing in HR-CESM but not LR-CESM.

## 4    Summary and Discussion

We analyzed the Community Earth System Model's Atlantic freshwater budget in a high resolution, strongly eddying and in a low resolution, non-eddying ocean component indicated here by HR-CESM and LR-CESM, respectively. We compared present day control simulations (CTRL) with observational data and analyzed changes under a climate change scenario with increasing greenhouse gases (RCP). Previous studies have analysed the Atlantic freshwater budget's present day state with strongly eddying ocean models (Treguier et al., 2012) or investigated the freshwater budget under climate change but with coarse resolution ocean models (Drijfhout et al., 2011), but this is the first analysis of the freshwater budget under climate change investigating the effect of strongly eddying oceans. Apart from the ocean horizontal resolution in the CESM, also the atmosphere model component version and resolution differ. However, the mean surface freshwater fluxes are comparable where ocean biases are comparable and the forced hydrological cycle response is similar between HR- and LR-CESM (Fig. 6). In validating the simulations, uncertainty in observations, particularly in the different $P - E$ products (Fig. 2), must be acknowledged (Trenberth et al., 2011). A multidecadal variability signal, significant with respect to a red noise null hypothesis, also exists in the HR-CESM simulation (Jüling et al., 2020). This could potentially influence the results, but the magnitude of the response to the strong RCP forcing is very large compared to this internal variability.

Increasing the resolution of the ocean component enables more realistic simulation of currents, eddies and overflows, and the circulation features such as the Gulf Stream separation or the Agulhas retroflection are better represented in the HR-CESM simulation (Fig. 4). We find that many ocean biases are reduced in HR-CESM compared to LR-CESM. Although the HR-CESM ocean presents more realistic boundary conditions to the atmosphere with more energy at smaller spatial and temporal scales (Kirtman et al., 2012), the atmosphere freshwater flux CTRL mean and RCP trends are similar between the two model setups (cf. Figs. 2d, 6, and 9). The large scale hydrological cycle strengthens similarly with generally warming surface temperatures, the exception being the cooling NA-SPG (Fig. 6). Also the AMOC weakens similarly (Fig. 5) in both RCP simulations, such that any differences in the simulated responses are likely due to the different ocean model resolution. The mean and trend of the ocean freshwater and salt transport, its convergence, and its decomposition differ between HR- and LR-CESM, especially in regions of strong eddy activity (cf. Figs. 7 and 9).

By comparing the CTRL simulations against observations relevant to the freshwater budget, we find that the HR-CESM biases are notably reduced compared to the LR-CESM setup. In particular, we diagnosed reduced biases of SST (Figs. 1 and A1), the precipitation minus evaporation fluxes (Figs. 2 and A2), the 3D Atlantic salinity distribution (Figs. 3 and A3). Two phenomena contribute to the strong meridional LR-CESM surface salinity bias gradient that also plagues other coarse resolution models (Mecking et al., 2017): first an unrealistically large import of too fresh surface waters from the Southwest

Indian Ocean into the South Atlantic (cf. Figs. 4 and A3), and second the southward shift of the ITCZ due to a more asymmetric Atlantic meridional SST bias (cf. Figs. 1 and 2). The structure and magnitude of the HR- and LR-CESM CTRL meridional freshwater transport terms (Fig. 7) are generally similar to those found in earlier studies with non-eddying models (Yin and Stouffer, 2007; Skliris et al., 2020) and eddying models (Mecking et al., 2016).

Despite similar atmospheric changes and AMOC slowdown, there are many notable differences between the HR- and LR-CESM simulations. Forced circulation changes differ in that the HR-CESM Gulf Stream moves north and the SPG circulation strength trends show a dipole pattern as opposed to a large-scale weakening in the LR-CESM simulation (Fig. 4). Also, Arctic surface freshwater fluxes change differently and the sea ice response may be underestimated due to low-biased heat transport into the Arctic in the LR-CESM simulation (Fig. 9). The large-scale structure of the $F_{tot}$ transport is similar and so is the forced response, with the exception of the STG-SPG boundary around 45°N where the LR-CESM shows no trends in any transport component, but the HR-CESM exhibits a large negative $F_{tot}$ trend, due in equal parts to the eddy and overturning components (Fig. 7). The decomposition between overturning and azonal components differs between the HR-CESM and LR-CESM simulations as the azonality of both the salinity and velocity fields differ. Eddy fluxes are significant at the northern and southern boundaries of the STGs (Treguier et al., 2012, 2014).

The evolution of the AMOC under climate change is of great interest and based on our results, and that of others, simulating strongly eddying oceans does not appear to systematically influence that response (Gent, 2018; Hirschi et al., 2020). The CTRL AMOC strength and reduction under the RCP scenario are almost identical between the simulations with a reduction of $\sim 5\,\mathrm{Sv}$ in 100 years from $18\,\mathrm{Sv}$ which compares well with the observed AMOC strength at the RAPID array at 26.5°N of $17.0\,\mathrm{Sv}$ (Smeed et al., 2018). The reduced heat transport by the AMOC into the subpolar gyre constitutes a positive atmospheric feedback in that evaporation is reduced, freshening the surface in the sinking regions. The salt-advection feedback is another positive AMOC feedback and it can lead to multiple equilibria if the overturning circulation exports freshwater from the Atlantic basin. A weakened AMOC would export less freshwater which would ultimately further suppress deep water formation in the North Atlantic and vice versa. As it is not possible to prove the existence of multiple AMOC equilibria with modern coupled climate models due to the high dimensionality and the prohibitive computational cost of equilibrating the ocean circulation after millennia, it is desirable to use scalar indicators based on simpler models. The import of freshwater to the Atlantic by the overturning circulation $F_{ovS}$ can hence provide further insight into the question of AMOC stability if atmospheric feedbacks are negligible (Huisman et al., 2010). Observations suggest a negative $F_{ovS}$ between $-0.28$ and $-0.05\,\mathrm{Sv}$ at present (Weijer et al., 2019). Due to their salinity bias at 34°S, both HR-CESM and LR-CESM CTRL simulations import freshwater into the Atlantic, but this bias is significantly reduced in HR-CESM (Figs. 3 and 10). This bias, from which all coarse resolution CMIP5 models suffer (Mecking et al., 2017; Gent, 2018), is countered by salinification of the surface under radiative forcing decreasing the $F_{ovS}$ value which indicates decreasing stability.

The ocean mean state as well as the forced response are different with higher resolution, but from our two RCP simulations we cannot discern any systematic effect on AMOC response to climate change as it is a large-scale flow feature and the correct simulation of the sinking regions is likely more important (Hirschi et al., 2020; Jackson et al., 2020). Yet due to the reduced salinity biases in particular at 34°S in HR-CESM, the indicator of the multiple equilibrium regime $F_{ovS}$ suggest that the salt-

advection feedback can destabilize the AMOC in the 21st century. However, the HR-CESM freshwater overturning transport response is meridionally incoherent and hence freshwater may not be simply advected northward with the AMOC. As the transport decomposition is further complicated by an eddy term, it is questionable whether the simple indicator is useful for quantifying the salt-advection feedback and it may have to be adapted. In the absence of eddy terms and changes in the salt

reservoirs, the overturning and azonal components must balance which was used by de Vries (2005) to change the sign of $F_{ovS}$. By changing the azonality of the freshwater surface fluxes at 34°S and hence the gyre transport, Cimatoribus et al. (2012) was able to collapse the AMOC without any further changes, suggesting that also this component of the transport must be taken into account when assessing the stability. Furthermore, any interpretation of the $F_{ovS}$ in short strongly eddy simulations should be undertaken with care. Figure 10 shows that the HR-CESM CTRL simulation switches sign from negative to positive $F_{ovS}$

only after 150 years as the ocean equilibrates.

To conclude, the changes in the Atlantic freshwater budget due to global warming are fairly robust to the resolution improvement from a diffusive to a strongly eddying ocean in CESM. This strengthens trust in using the current generations of coupled climate models (CMIP5, CMIP6) and their AMOC change projections, which are computationally significantly cheaper to perform. On the other hand, the biases in the present-day state are strongly reduced in the strongly eddying ocean version of

CESM and hence indicate that better parameterizations are needed in the CMIP5/CMIP6 models to reduce these biases. This reduction can be crucial for assessing the probability of tipping of the AMOC under future climate change.

## Appendix A:  Additional Model-Observation comparison

In this appendix, we present global maps of the model-observation comparisons of section 2.2.

Regarding the SST bias world map (Fig. A1), in the HR-CESM CTRL simulation warm biases are located in the high

latitudes and the tropical Indo-Pacific and cold biases in the Indo-Pacific subtropical gyres and the tropical and subtropical Atlantic. The LR-CESM CTRL simulation SST bias is more asymmetric about the equator with large-scale cold biases only in the Northern Hemisphere subtropical gyres and the southern edge of the NA-SPG.

## Appendix B:  Budget calculation

As Schauer and Losch (2019) point out, the values of the freshwater flux terms are non-linearly dependent on the chosen

reference salinity $S_0$. Traditionally, the AMOC bistability question with respect to the salt advection feedback has been framed in terms of freshwater import due to the meridional overturning.

### B1  Freshwater budget

We define freshwater fluxes in the ocean relative to a reference salinity of $S_0 = 35$ in units of $1\,\mathrm{Sv} = 10^6\,\mathrm{m^3\,s^{-1}}$. The freshwater flux budget of an arbitrary full depth ocean volume is given by Eq. (1) which we repeat here for a self-contained presentation

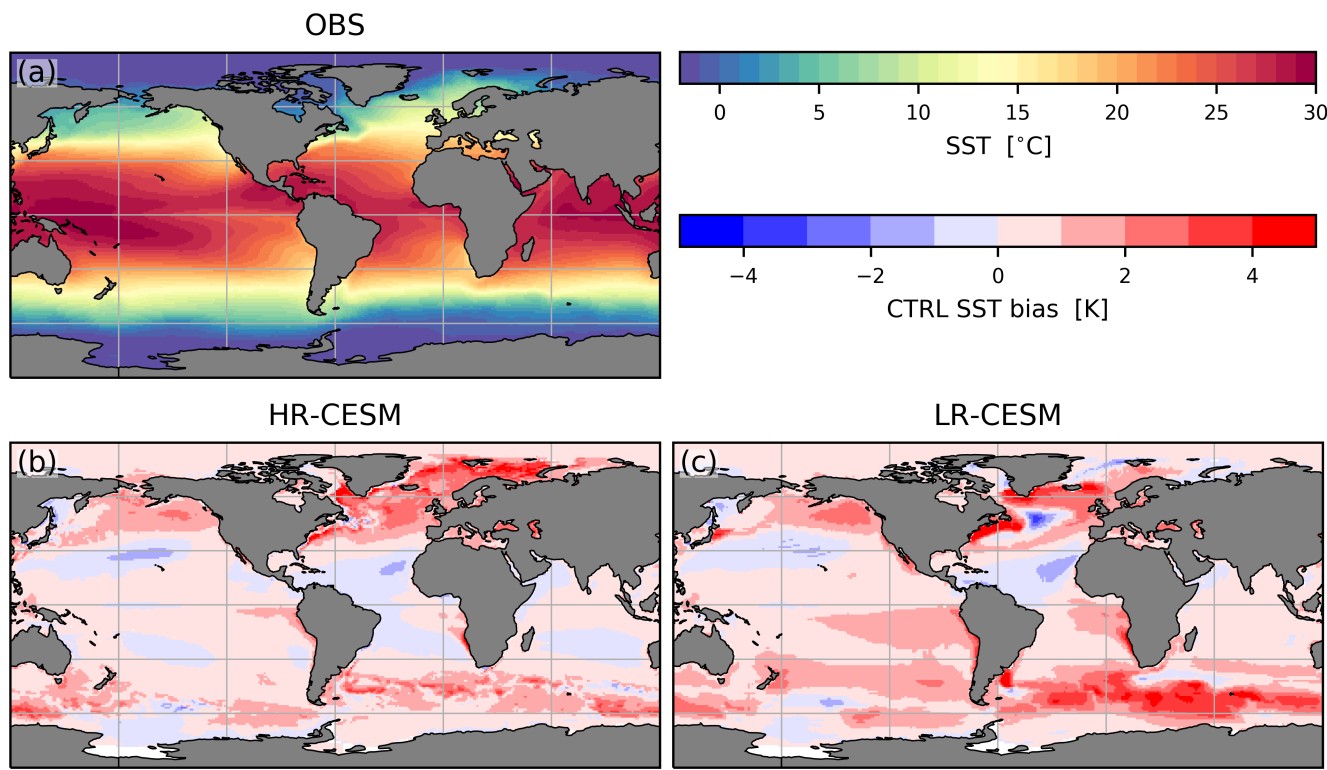

**Figure A1.** Bias of annual SSTs of the HR-CESM (left) and LR-CESM (right) CTRL simulations, like Fig. 1.

of the budget calculations:

$$\frac{\mathrm{d}\bar{W}}{\mathrm{d}t} = F_\nabla + F_{surf} + F_{mix}$$

where $\bar{W} = -\frac{1}{S_0} \int \int \int S - S_0 \, \mathrm{d}V$ is the freshwater content of the volume $V$. The first term on the right hand side, $F_\nabla$, is due to the advection of freshwater gradients across the vertical boundary $b$, which is full depth and encloses the volume $V$:

$$F_\nabla = \int \int \boldsymbol{u} \cdot \nabla W \, \mathrm{d}b / \int \int \mathrm{d}b \qquad \text{(B1)}$$

The second term is the freshwater flux at the surface comprising precipitation $P$, evaporation $E$, runoff from land $R$ and ice $I$, as well as sea ice melt $M$ (brine rejection $B$) which are all defined as positive (negative) freshwater fluxes into the ocean:

$$F_{surf} = P + E + R + I + M + B \qquad \text{(B2)}$$

The last term, $F_{mix}$, captures diffusion (including eddy parametrizations), errors introduced by the time averaging of the output and the choice of the reference salinity $S_0$ and is calculated as a residual:

$$F_{mix} = \frac{\mathrm{d}\bar{W}}{\mathrm{d}t} - F_{surf} - F_\nabla \qquad \text{(B3)}$$

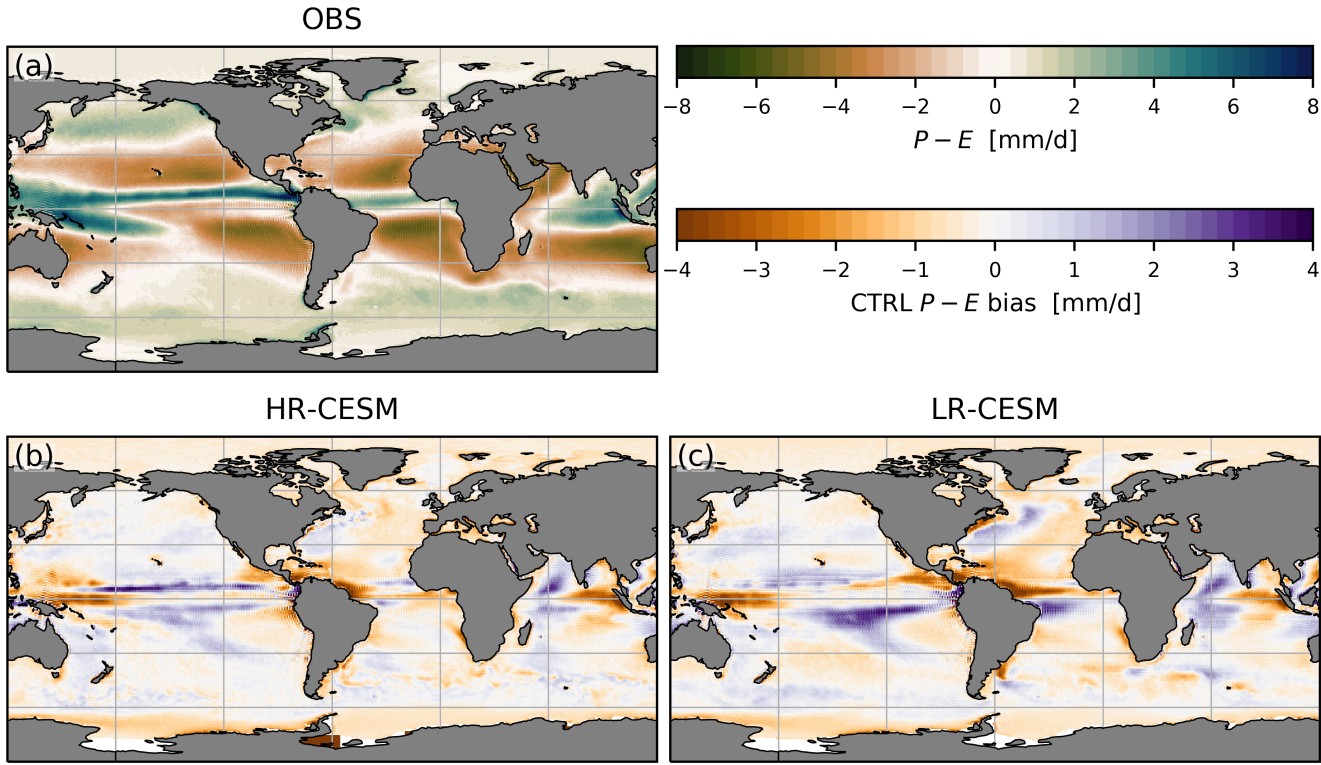

**Figure A2.** Comparing HR-CESM (left) and LR-CESM (right) precipitation to ERA-Interim, like Fig. 2.

Furthermore, we ignore changes in dynamic sea level in the calculation of $\bar{W}$ such that these small effects are included in $F_{mix}$.

To ascertain whether a perturbation in the overturning is amplified or damped through the salt advection feedback, the freshwater transport due to the overturning is evaluated at the southern boundary ($F_{ovS}$). In general, the advective term can be divided into a barotropic component $F_{bt}$, an overturning component $F_{ov}$, an azonal component due to the gyre circulation $F_{az}$,

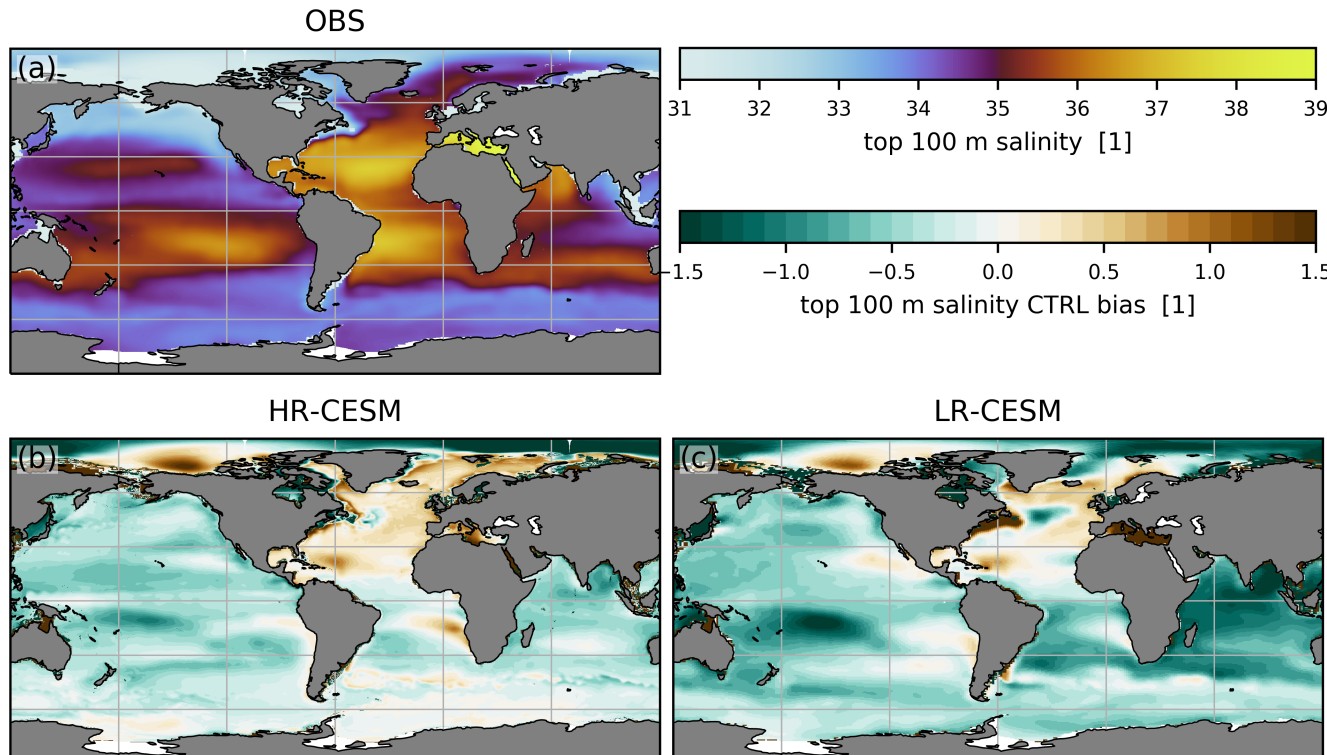

**Figure A3.** The salinity bias of the upper $100\,\mathrm{m}$, like Fig. 3.

and an eddy component $F_{eddy}$ such that:

$$F_{\nabla} = (F_{bt} + F_{ov} + F_{az} + F_{eddy})\big|_{y=\theta_S}^{\theta_N} \tag{B4}$$

$$F_{bt}(y) = -\hat{v}\frac{\hat{S} - S_0}{S_0} \tag{B5}$$

$$F_{ov}(y) = -\frac{1}{S_0}\int \left[\int_W^E v^*\,\mathrm{d}x\right][\langle S \rangle - S_0]\,\mathrm{d}z \tag{B6}$$

$$F_{az}(y) = -\frac{1}{S_0}\int_{-H}^{z}\int_W^E v'S'\,\mathrm{d}x\,\mathrm{d}z \tag{B7}$$

$$F_{eddy}(y) = -\frac{1}{S_0}\int_{-H}^{z}\int_W^E \left(\overline{v\,[S - S_0]} - \overline{v}\times\overline{S}\right)\mathrm{d}x\,\mathrm{d}z \tag{B8}$$

where $F_{\nabla}$ is evaluated between the Southern and Northern boundaries, $\theta_{S/N}$. The hat notation $\hat{q}$ of an arbitrary quantity $q$ denotes the section average, $\hat{q} = \int\int q\,\mathrm{d}x\,\mathrm{d}z / \int\int \mathrm{d}x\,\mathrm{d}z$. In case $q = v$, $\hat{v}$ is the barotropic velocity and $v^* = v - \hat{v}$ is the baroclinic


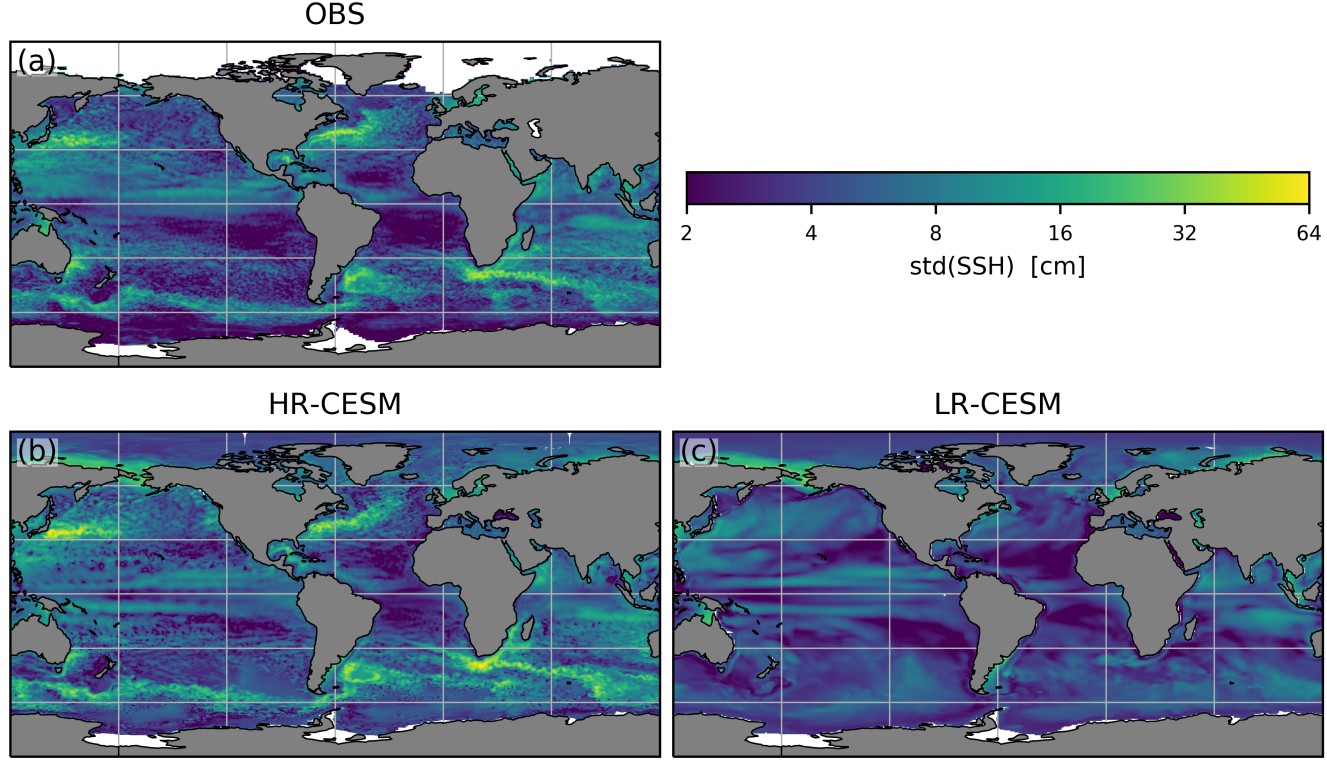

**Figure A4.** The standard deviation of the observed sea surface height (a) and the modeles dynamic sea level (b/c), like Fig.4.

velocity. Angled brackets $\langle q \rangle = \int q \, \mathrm{d}x / \int \mathrm{d}x$ denote zonal averaging, while primed quantities $q' = q - \langle q \rangle$ are deviations from zonal means.

In specific case of the Atlantic-Arctic freshwater budget, the oceanic advection term can be decomposed into advective freshwater fluxes at a Southern and Northern boundary, usually at the latitudes of Cape Agulhas at 34°S and Bering Strait at 68°N ($F_{BS}$), plus the the Mediterranean inflow ($F_{Med}$).

$$F_{Med} = -\frac{1}{S_0} \int_{-H}^{z} \int_{\theta_{Med,S}}^{\theta_{Med,N}} \overline{u(S - S_0)}(x = 5.5°W) \, \mathrm{d}z \, \mathrm{d}y \tag{B9}$$

where $u$ is the zonal velocity.

Note that sometimes $F_{ov}$ is defined to include the barotropic component (e.g. de Vries (2005)):

$$F_{ov} = -\frac{1}{S_0} \int \bar{v} \left[ \langle S \rangle - S_0 \right] \, \mathrm{d}z \tag{B10}$$

Equations (B6) and (B10) are equal if the reference salinity is equal to the section average, $S_0 = \hat{S}$, and the barotropic transport is zero, $F_{bt} = 0$.

Due to the volume-conserving, virtual salt flux formulation of the ocean model, the barotropic meridional volume transport throughout the Atlantic equals that through Bering Strait (Table 2). The barotropic freshwater transport thus depends only on the section average salinity which is so close to the reference salinity, $S_0 \approx \hat{S}$, that $F_{bt}$ is negligibly small compared to the other transport components and hence not shown.

## B2  Eddy-mean decomposition

To calculate the eddy terms, we use the eddy-mean decomposition of the total flux:

$$\overline{xy} = \bar{x}\bar{y} + \overline{x'y'} \tag{B11}$$

where the overbar $\bar{x}$ denotes a time average, which we choose to be annual so as to include seasonality effects from the eddy term (Fig. B1 compares an annual with a monthly cutoff time scale), and primed quantities $x'$ denote eddy terms.

Neither the total nor the eddy freshwater transport terms, $\overline{v[S - S_0]}$ and $-\frac{1}{S_0}\overline{v'[S - S_0]'}$, are part of the model output.
However, the total salt transport $\overline{vS}$ is, so that one can calculate the eddy salt transport:

$$F_{eddy}^S = \int\limits_{-H}^{z}\int\limits_{W}^{E} \overline{vS} - \overline{v} \times \overline{S}\,\mathrm{d}x\,\mathrm{d}z \tag{B12}$$

The freshwater eddy transport is linearly related to the eddy salt transport

$$F_{eddy} = -\frac{1}{S_0}\overline{v'[S - S_0]'} = -\frac{1}{S_0}\overline{v'S'} = -\frac{1}{S_0}F_{eddy}^S \tag{B13}$$

and the total freshwater flux is thus:

$$F_{total} = F_{bt} + F_{ov} + F_{az} + F_{eddy} \tag{B14}$$

Figure B1 shows the effect of the cutoff time scale on all transport terms. Only in HR-CESM is the eddy term of relevant magnitude compared to the mean terms. Naturally, the total transport is unaffected, while the mean terms, in particular the azonal term, gain in strength at the expense of the eddy term. The HR-CESM eddy term (green line in Fig. B1a) is reduced to roughly half its strength in the ITCZ and the NA-STG while it becomes negligible in the SA-STG. Under climate change,
trends in eddy transports are similarly reduced at a monthly cutoff time scale, suggesting that much of the changes in the annual eddy trends are driven by a changing seasonality.

The LR-CESM configuration diagnoses the bolus eddy induced advection (from the GM parametrization) and the submesoscale advection (from the biharmonic diffusion). Figure B2 compares the diagnosed GM (blue) and submesoscale (orange) freshwater transport terms to the eddy term (Eq. B13; green). The GM parameterization term is generally significantly smaller
than the term arising from the Eulerian eddy-mean decomposition (for both monthly and annual cutoff time scales), but can be comparable in size (e.g. at the equator or the NA-STG). The submesoscale term, on the other hand, is negligible compared to the other two terms.

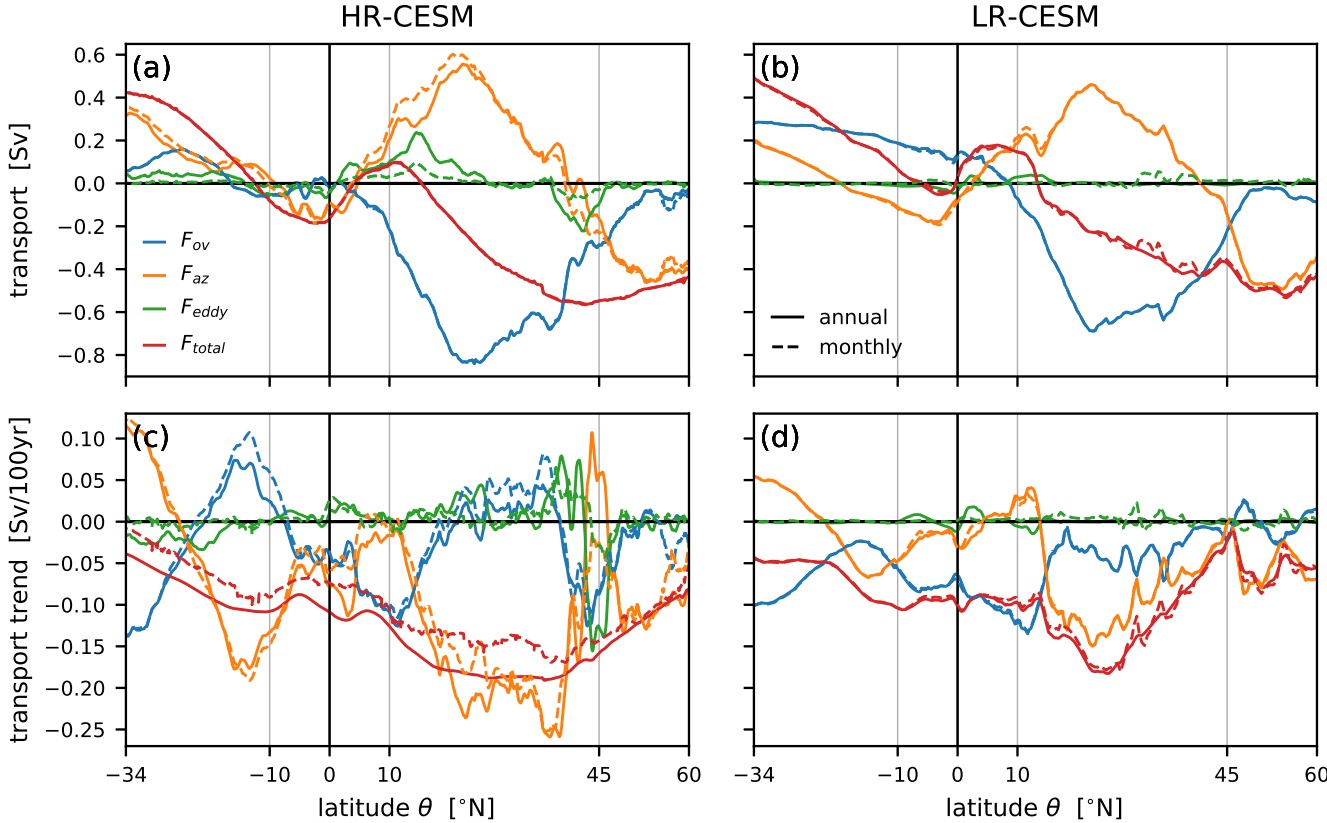

**Figure B1.** Freshwater transport terms calculated from annually (solid lines; as in Figs. 7, 9, and 10) vs. monthly (dashed) averaged model output. The difference is the effect of seasonality.

*Code and data availability.* The analysis scripts are available at *doi.org/10.5281/zenodo.4537845*, while the model output is stored at SURFsara and available upon request to the corresponding author.

*Author contributions.* AJ, AvdH, DC, and HD conceived the presented ideas in this study. AJ performed the analysis and wrote the manuscript. XZ carried out some initial analysis. AvdH, DC and HD contributed to writing the paper.

*Competing interests.* The authors declare no competing interests.

*Acknowledgements.* This work was carried out under the program of the Netherlands Earth System Science Centre (NESSC), financially supported by the Ministry of Education, Culture and Science (OCW) (Grantnr. 024.002.001). AJ has been fully supported by NESSC, AvdH

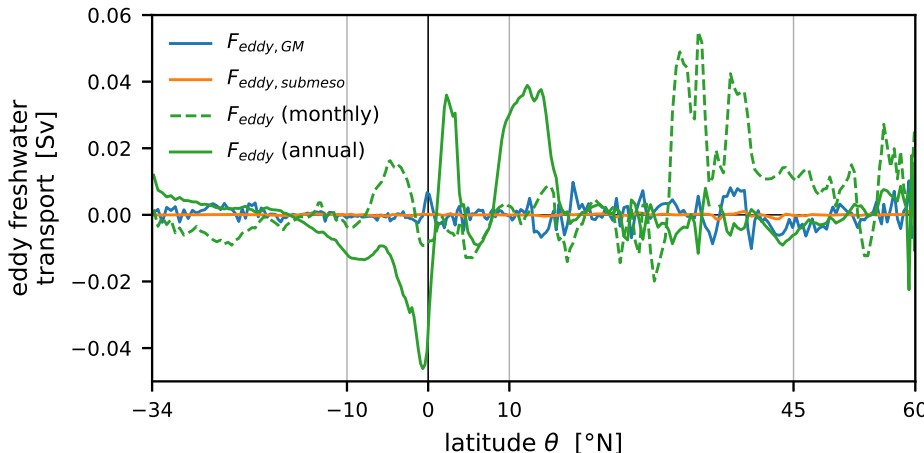

**Figure B2.** Freshwater transport terms calculated from the eddy salt advection terms diagnosed by the LR-CESM model compared to the monthly and annual $F_{eddy}$ term (green) which are shown in Figs. 7b and B1. The model diagnoses salt advection by the GM parametrization (blue) and the submesoscale mixing parameterization (orange).

and HD in part by NESSC. The computations were performed on the Cartesius high performance computer at SURFsara in Amsterdam. Use of the Cartesius computing facilities was sponsored by the Netherlands Science Foundation (NWO) under the project 17239. We thank Michael Kliphuis (IMAU) for carrying out the CESM simulations.

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
