# Peer review of "The Atlantic's Freshwater Budget under Climate Change in the Community Earth System Model with Strongly Eddying Oceans"

_Ocean Science, 2020_

## Referee Comment (RC1) · Anonymous Referee #1 · 18 Sep 2020

Review of os-2020-76 – The Atlantic's Freshwater Budget under Climate Change in the Community Earth System Model with Strongly Eddying Oceans

This manuscript gives a very detailed description of the components of the Atlantic Freshwater budget and how it changes under future climate projections. The study uses two versions of the CESM with different resolutions, one which is strongly eddying and the other which parametrizes eddies in the ocean. The results show that the higher resolution model has smaller biases than the lower resolution model. The authors also show that there isn't much difference in the response of the AMOC to the future $CO_2$ projections between the high and low resolution simulations. While there are aspects

of the freshwater budget between the high and low resolution simulations that similar the high resolution simulation has the additional transport due to eddies at the gyre boundaries.

I feel that the detail of the freshwater budget presented in this study is of interest to the scientific community. However, while I feel that the scientific content is sound and contents is well organized, I believe there is quite a bit of room for improvement to this study.

Major Comments:

1. Throughout the entire study references to existing literature is a bit on the sparse side. For instance, the second paragraph of the introduction only has one reference. References where values based on observations/models are stated should be included (e.g. line 33 – Frajka-Williams et al. 2019 (or Smeed et al. 2018) for AMOC strength or line 7 – Woodgate and Aagaard 2005 for Bering Strait through flow). Another example is when discussing the freshwater budget comparing how the results in the paper support or contradict other previous studies (i.e. Skliris et al. 2020 in their figure 11 do something very comparable to the manuscript's figure 9, Similarly, the studies Yin and Stouffer 2007 and Mecking et al. 2016 also do freshwater budgets).

2. A major difference between the high and low resolution simulations is the model's ability to handle eddies. Therefore, it would be nice to see a figure showing the differences in the eddy activity between the HIGH and LOW models (e.g. something similar to Delworth et al. 2012 their figure 14).

3. The short names HIGH and LOW used through the manuscript when they aren't followed by the word simulation makes a quick read of some sentences confusing (e.g. line 158,178,213,etc).

4. It would be very helpful to have some of the figures include extra panels (redone to show anomalies (differences between CTRL and RCP) as opposed to just the absolute

values. In particular, this would be very helpful for Figures 7 and 9, making it a lot clearer what the changes in the future climate projections are. The text would probably benefit from a few small tweets to reflect these figures this as well.

5. For several reasons I find Figure 9 nice but at the same time too complex. I really like comparisons of budgets using bar charts since the make it easy to see the relative differences between the different components. Panel a is good but panel b contains too much information. There is a lot of information on panel b, perhaps breaking panel b down into several panels will make it simpler, e.g. have a panel of surface flux break down and another for advection break down. The axis on the bottom of panel b are also confusing, first of all that they are different sizes for horizal and vertical and where do the units, Sv/100yr come into play? Also, the arrows on the end of the bars are difficult to see differences between the HIGH and LOW simulations. It would be helpful to make an anomaly version of this figure.

Minor Comments:

1. At a few points in the manuscript a freshwater and salinity budget is mentioned, but only the freshwater budget is discussed. Even though they are quite similar it is probably worth just mentioning freshwater budget. (e.g. line 1 and other places)

2. Line 26 – Mecking et al. 2016 also showed this

3. Lines 32 and 37 – Do you mean approximately instead of some?

4. Line 44/45 – What about heat flux changes? i.e. Gregory et al. 2005

5. Lines 51-66 – The references Weaver et al. 2012 and Liu et al. 2014 are also quite relevant for this paragraph.

6. Line 66 – reference? Is this line even needed?

7. Line 90 – You are comparing ocean-only (Deshayes et al. 2013) to coupled simulations (Mecking et al. 2017). The ocean only simulations use salinity restoring which

is potentially the reason why the ocean only simulations have a negative FovS as opposed to the difference in resolution.

8. Line 113/114 – What impact does only using CO2 have?

9. Line 139 – reference for HadISST missing

10. Line 141-142 – Is there a warming trend in the beginning of CTRL before the reference period for this study is taken? Since the reference period of HadISST is 10 years before 2000 and 20 years after, I would naively not expect a warming.

11. In some 2D figures there is a line of missing data for the LOW simulations (e.g. Figures 1,3c,i,4b,6).

12. Line 148 – Refence to indicate that the warming hole response is expected i.e. Drijfhout et al. 2012

13. Line 162/Figure 2e – The integration of the surface fresh water fluxes is typically done North to South because that's the direction of the barotropic flow through the Atlantic (i.e. Skliris et al. 2020 Figure 6, Mecking et al. 2017 Figure 6), this way it will line up to the difference between inflow into the Bering Strait and outflow at 34S

14. Table 2, it would be nice to also include the transports at 34S (or as close to it as possible, see Bryden et al. 2011 for an observational estimate)

15. Table 2 – how is salt transport defined and how does it relate to freshwater transport?

16. Line 180 – I don't understand the barotropic streamfunction computation, the constant of integration should have an x/longitude dependence and not along the coast of the Atlantic side of Africa

17. Line 211 – By colours do you mean shading?

18. Line 221 – Refence for AMOC strength

19. Line 229 – The freshwater loss seems large (i.e. larger than CMIP5, see Skliris et al. 2020 Figure 6)

20. Line 234 – Which linear trends are being referred to here?

21. Figure 7 – why is 36N masked?

22. Figure 7 – Do freshwater transports through the Bering Strait and Strait of Gibraltar change?

23. Figure 7 and 10, most obvious in Figure 7, the thick and thin line thicknesses are very difficult to distinguish.

24. Line 256 (some other places later) – it would be nice to reference earlier figures/figure panels at the end of sentences to help make connections i.e. line 256 – Fig. 6k/l and line 266 Fig. 7

25. Figure 8 – Is it worth including 34S section and/or Atlantic zonal meam?

26. Figure 8 – the freshening trend in the SPG is in line with the warming hole that goes along with a weakening AMOC (i.e. Menary et al. 2018, Fig. 7) – should be mentioned in the text below

27. Line 305 – Any idea why there are these differences in salinity?

28. Line 321 – durface should be surface

29. Line 361 – Also in Liu et al. 2014

Frajka-Williams, E., Ansorge, I.J., Baehr, J., Bryden, H.L., Chidichimo, M.P., Cunningham, S.A., Danabasoglu, G., Dong, S., Donohue, K.A., Elipot, S. and Heimbach, P., 2019. Atlantic meridional overturning circulation: Observed transport and variability. Frontiers in Marine Science, 6, p.260.

Smeed, D.A., Josey, S.A., Beaulieu, C., Johns, W.E., Moat, B.I., Frajka‐Williams, E., Rayner, D., Meinen, C.S., Baringer, M.O., Bryden, H.L. and McCarthy, G.D., 2018.

The North Atlantic Ocean is in a state of reduced overturning. Geophysical Research Letters, 45(3), pp.1527-1533.

Woodgate, R.A. and Aagaard, K., 2005. Revising the Bering Strait freshwater flux into the Arctic Ocean. Geophysical Research Letters, 32(2).

Skliris, N., Marsh, R., Mecking, J. and Zika, J.D., 2020. Changing water cycle and freshwater transports in the Atlantic Ocean in observations and CMIP5 models. Climate Dynamics.

Yin, J. and Stouffer, R.J., 2007. Comparison of the stability of the Atlantic thermohaline circulation in two coupled atmosphere–ocean general circulation models. Journal of climate, 20(17), pp.4293-4315.

Mecking, J.V., Drijfhout, S.S., Jackson, L.C. and Graham, T., 2016. Stable AMOC off state in an eddy-permitting coupled climate model. Climate Dynamics, 47(7-8), pp.2455-2470.

Delworth, T.L., Rosati, A., Anderson, W., Adcroft, A.J., Balaji, V., Benson, R., Dixon, K., Griffies, S.M., Lee, H.C., Pacanowski, R.C. and Vecchi, G.A., 2012. Simulated climate and climate change in the GFDL CM2. 5 high-resolution coupled climate model. Journal of Climate, 25(8), pp.2755-2781.

Gregory, J.M., Dixon, K.W., Stouffer, R.J., Weaver, A.J., Driesschaert, E., Eby, M., Fichefet, T., Hasumi, H., Hu, A., Jungclaus, J.H. and Kamenkovich, I.V., 2005. A model intercomparison of changes in the Atlantic thermohaline circulation in response to increasing atmospheric CO2 concentration. Geophysical Research Letters, 32(12).

Weaver, A.J., Sedláček, J., Eby, M., Alexander, K., Crespin, E., Fichefet, T., Philipponâ ̆ŘBerthier, G., Joos, F., Kawamiya, M., Matsumoto, K. and Steinacher, M., 2012. Stability of the Atlantic meridional overturning circulation: A model intercomparison. Geophysical Research Letters, 39(20).

Liu, W., Liu, Z. and Brady, E.C., 2014. Why is the AMOC monostable in coupled general

circulation models?. Journal of climate, 27(6), pp.2427-2443.

Drijfhout, S., Van Oldenborgh, G.J. and Cimatoribus, A., 2012. Is a decline of AMOC causing the warming hole above the North Atlantic in observed and modeled warming patterns?. Journal of Climate, 25(24), pp.8373-8379.

Mecking, J.V., Drijfhout, S.S., Jackson, L.C. and Andrews, M.B., 2017. The effect of model bias on Atlantic freshwater transport and implications for AMOC bi-stability. Tellus A: Dynamic Meteorology and Oceanography, 69(1), p.1299910.

Bryden, H.L., King, B.A. and McCarthy, G.D., 2011. South Atlantic overturning circulation at 24 S. Journal of Marine Research, 69(1), pp.38-55.

Menary, M.B. and Wood, R.A., 2018. An anatomy of the projected North Atlantic warming hole in CMIP5 models. Climate dynamics, 50(7-8), pp.3063-3080.

---

## Referee Comment (RC2) · Anonymous Referee #2 · 18 Oct 2020

The manuscript "The Atlantic's Freshwater Budget under Climate Change in the Community Earth System Model with Strongly Eddying Oceans" by Jüling, Zhang, Castellana, von der Heydt, and Dijkstra provides a detailed analysis of the salt/freshwater budget of the (North) Atlantic and the role of mesoscale eddies in meridional transport and changes thereof under global warming. This is a very thorough study also validating the importance of explicitly resolving mesoscale eddies in global ocean/climate simulations and estimates of the bistability of the AMOC. The analysis and results are well embedder din existing literature and thus are an important contribution to the ongoing discussion on AMOC stability and eddy-resolving ocean simulations.

[Figure]

I recommend publication of the manuscript after considering the following minor comments.

MINOR COMMENTS (by line):

Title: For most parts of the paper the discussion focusses on the salt/freshwater budget of the North Atlantic and only little analysis and information is provided for the South Atlantic and its import pathways through Drake passage and Agulhas leakage. I thus suggest to add "North" to the title: "The North Atlantic's Freshwater . . ."

1f Please add specific ocean grid resolution information to the abstract: "We investigate the freshwater and salinity budget of the Atlantic and Arctic oceans in two configurations of the Community Earth System Model (CESM), one with a strongly eddying ocean on a 0.1ËŽ grid and one of coarser, non-eddying resolution (1.0ËŽ) typical of CMIP6 models."

27 "salt-advection feedback" should have a reference, e.g. Peltier and Vettoretti (2014)? [add. references provided below]

28f This sentence could use a reference as well, for example Behrens et al. (2013)

33f "17 Sv at 26.5ËŽN" Is this based on observations, e.g. RAPID, or your model simulations? Please add reference, for RAPID: Smeed et al. (2016) or Smeed et al. (2018), the latter already used in the manuscript at a later point (line 423). Should be cited here as well.

37 "0.8 Sv of relatively fresh Pacific" reference? For example Woodgate and Aargaard (2005)

39f "Freshwater is also exchanged with the Mediterranean Sea which is strongly evaporative." I would rather term this a salinity exchange, because Mediterranean outflow is very salty, i.e. the Atlantic provides "freshwater", which is in this terminology somewhat awkward.

42 "... and advect salt meridionally when there is a zonal salinity gradient." In the same sentence it is said that the gyres are wind-driven. This part sounds like they are driven by a zonal salinity gradient. I suggest to rephrase this part: "... and advect any zonal salinity gradient also in meridional direction." In context with the previous sentence, this most importantly means that the salinity differences caused by precipitation patterns in the ITCZ are advected poleward by the gyres. Maybe this should be stressed more.

43 "under greenhouse gas increases" rather is "under increasing greenhouse gas concentrations"

58-66 very nice, brief explanation of the impact of freshwater import from the south on the salt-advection feedback. However, in principle the AMOC does not import freshwater to the North Atlantic but rather negative salinity anomalies (in models often a virtual salt flux anyways, see line 105). Also, a note on the calculation of F_ovS would be helpful, i.e. the typical reference salinity and whether zonally averaged velocities are used (AMOC streamfunction) or actual transports in 3-D are computed—is there a standard in place already?

117 How does the difference in vertical ocean grid resolution (42 vs. 60 levels) affect overflows in the North Atlantic? In particular resolution at Denmark Strait has the potential to significantly affect the AMOC.

132 more precise: "Green lines in Figure 1b,c mark the bounding latitudes which . . ."

165 Meaning of this introductory sentence not quite clear. Circulation changes between models must affect much more than just Bering Strait exchange and Mediterranean outflow. Maybe simply drop this sentence? Or move to line 199.

174f please provide depth in meters (not km)!

174 The reference should more clearly point to Figure 3d for the northward extent of AAIW, which by the way seems not to reach 20ËŽN—maybe 10ËŽN—as stated here, and Figure 3j for the low salinity signature of AAIW.

176 Figure 3i shows a section at 34ËŽS and thus cannot serve as a reference for the addressed bias at 15-30ËŽN.

191 add "modelled" in ". . . this is the salinity of modelled North Atlantic . . ."

227 I assume brine rejection is counted as negative freshwater flux into the ocean; add parentheses: ". . . sea ice melt (and brine rejection) . . . defined as positive (negative) freshwater fluxes . . ."

244, 284ff and Appendix B: It is not quite clear to me whether your method of computing eddy transports accounts for eddy induced velocities from the GM parameterization, which I believe is used in the LOW model run. While it is quite obvious that the velocity field of LOW is much smoother than the one in HIGH (as you point out for Figures 5d and 5e), the unresolved eddy fluxes should partly be compensated by the eddy mixing scheme (GM as noted only later in line 289), which would provide eddy induced velocities. These should be included in the eddy transport discussed with Figure 7. In this respect, the comment on line 280f should be moved upward and included in the introduction to section 3.3. [I was a bit impatient when reading this page and would have preferred to read the discussion of lines 280ff earlier. However, when reading this page again, I now think the structure is OK only that a small comment in line 259 would help, such as "In the following we take a closer look at these two differences."

248 Providing the year 2100 value of the linear trend seems a good way to limit the effect of internal variability in illustrating the changes under RCP scenario. However, it would be helpful to also note the correlation and significance of the linear trends in the text (or a table?) as a goodness-of-fit affirmation.

257 The obvious difference between the green CTRL lines (F_eddy) of HIGH and LOW runs should be pointed out first before discussing the deviations in trends, i.e. that LOW does not "exhibit the negative transport trend" at the SPG-STG boundary.

272 here or earlier: reference Yang et al. for shift in ocean gyres [see full reference

below]

275 It could be noted in addition to the present discussion that the differences between LOW and HIGH are not only due to resolving eddies but also due to a generally better representation of boundary currents (azonal flow system) in HIGH, which I assume is the case.

280f this statement should be made earlier (see comments on line 244 and App.B).

343 It would be helpful if you indicate the bar color you are referring to. For the 32% vs 18% increase I assume you mean the total, i.e. red bars. This is a very nice, detailed analysis. And I also like Figure 9 very much but it is somehow difficult to keep track of the bars (colors) each sentience refers to. Adding hints for the color would help to link text and figure.

345 If sea ice is one of the few bigger differences between HIGH and LOW worth noting, then please add this flux to Figure 9! A "(not shown)" is not very satisfying here.

363 I cannot see the advantage of presenting the spin up timeseries for this discussion. (see more comments on Figure 10)

391 one reference to Jüling et al 2020 is sufficient in this line. Also, this citation lacks a journal and DOI in the reference list. Is the paper accepted/published already?

Tables

Table 1 In addition to the start year of the RCP run, please provide the length of the CTRL run (spinup?). Otherwise it looks like the high-resolution model was only spun up for 200 years, which would be very short to study the AMOC. Did these runs branch off of any longer CESM spinup?

Table 2 I recommend to use sign + for transports into the Atlantic-Arctic basin. (I would think the Mediterranean provides a net (virtual) salt inflow to the system, i.e. +1.2 and -0.032.)

Figures

Figure 1: please use fewer colors for all plots to enhances visibility of actual differences.

Figure 5: the offset applied to the linear trend lines is not necessary, I think. Just use tone down the color a little bit, then it can be placed right in top of the smoothed timeseries without information loss.

Figure 7: thick and thin lines of CTRL and RCP are barely distinguishable. Please increase the difference in thickness or use dashed lines for RCP.

Figure 10: I depreciate the change in timescale. I think this gives a wrong impression on the trend under RCP w.r.t. the equilibration process. Also, I cannot grasp the purpose of showing the spinup period at all. Why do you not focus on the last 100 years in both cases? Further, I suggest to add labels for the regimes defined by the sign on the y-axis, e.g. on the righthand side y-axis.

Appendix B: Does the velocity v in your calculations for the LOW model include eddy induced velocity components from an eddy parameterization such as GM? I think this should be included for a fair comparison between LOW and HIGH. The salinity distribution will inherently include the effect of such parameterization but does v?

511: Why annual mean? Since you have mean(vS) from model output you can compute the eddy transport also on monthly basis and thus eliminate the effect of seasonal variability from you calculation.

Additional References

Behrens, E. , A. Biastoch, and C.W. Böning (2013), Spurious AMOC trends in global ocean sea-ice models related to subarctic freshwater forcing, Ocean Modelling (69), 39-49, doi:10.1016/j.ocemod.2013.05.004.

Peltier, W. R., and G. Vettoretti (2014), Dansgaard-Oeschger oscillations predicted in a comprehensive model of glacial climate: A "kicked" salt oscillator in the Atlantic,

Geophys. Res. Lett., 41, doi:10.1002/2014GL061413.

Smeed, D., McCarthy, G., Rayner, D., Moat, B.I., Johns, W.E., Baringer, M.O., Meinen, C. S., 2016. Atlantic meridional overturning circulation observed by the RAPID-MOCHA-WBTS (RAPID-meridional overturning circulation and heatflux array-wes- tern boundary time series) array at 26 N from 2004 to 2015. doi:10.5285/35784047- 9B82-2160-E053-6C86ABC0C91B.

Smeed, D. A., Josey, S. A., Beaulieu, C., Johns, W. E., Moat, B. I., Frajka-Williams, E., et al. (2018). The North Atlantic Ocean is in a state of reduced overturning. Geophysical Research Letters, 45, 1527–1533. https://doi.org/10.1002/ 2017GL076350

Woodgate, R.A., Aagaard, K., 2005. Revising the Bering Strait freshwater flux into the Arctic Ocean. Geophys. Res. Lett. 32. http://dx.doi.org/10.1029/2004GL021747.

Yang, H., Lohmann, G., Krebs-Kanzow, U., Ionita, M., Shi, X., Sidorenko, D., et al. (2020). Poleward shift of the major ocean gyres detected in a warming climate. Geophysical Research Letters, 47, e2019GL085868, doi:10.1029/2019GL085868.

---

## Author Comment (AC2) · 27 Nov 2020

DOI: 10.5194/os-2020-76
Version: Revision
Title: The Atlantic's Freshwater Budget under Climate Change in the Community Earth System Model
Authors: André Jüling, Xun Zhang, Daniele Castellana, Anna S. von der Heydt, Henk A. Dijkstra

Point by point reply to reviewer #2

November 27, 2020

We thank the reviewer for their careful reading and for the useful comments on the manuscript. We marked the points we have already addressed with ticks.

**1 Reviewer Summary:**

The manuscript "The Atlantic's Freshwater Budget under Climate Change in the Community Earth System Model with Strongly Eddying Oceans" by Jüling, Zhang, Castellana, von der Heydt, and Dijkstra provides a detailed analysis of the salt/freshwater budget of the (North) Atlantic and the role of mesoscale eddies in meridional transport and changes thereof under global warming. This is a very thorough study also validating the importance of explicitly resolving mesoscale eddies in global ocean/climate simulations and estimates of the bistability of the AMOC. The analysis and results are well embedded in existing literature and thus are an important contribution to the on-going discussion on AMOC stability and eddyresolving ocean simulations. I recommend publication of the manuscript after considering the following minor comments.

**2 Minor Comments:**

 ✓ Title: For most parts of the paper the discussion focusses on the salt/freshwater budget of the North Atlantic and only little analysis and information is provided for the South Atlantic and its import pathways through Drake passage and Agulhas leakage. I thus suggest to add "North" to the title: "The North Atlantic's Freshwater ..."

(Title) While we see the merits of the argument, we believe we should keep the more general title because we focus on the import of freshwater at 34°S, maps are provided for the whole Atlantic+Arctic, and we analyze the transport and flux terms also south of the equator.

- 2. ✓ 1f Please add specific ocean grid resolution information to the abstract: "We investigate the freshwater and salinity budget of the Atlantic and Arctic oceans in two configurations of the Community Earth System Model (CESM), one with a strongly eddying ocean on a 0.1° grid and one of coarser, non-eddying resolution (1.0°) typical of CMIP6 models. "

  (1.1) we added the grid spacing to the abstract.
- 3. ✓ 27 "salt-advection feedback" should have a reference, e.g. Peltier and Vettoretti (2014)? [add. references provided below]
  (1. 27) reference added.

- 4. ✓ 28f This sentence could use a reference as well, for example Behrens et al. (2013)
  (1. 28) added reference.
- 5. ✓ 33f "17 Sv at 26.5° N" Is this based on observations, e.g. RAPID, or your model simulations? Please add reference, for RAPID: Moat et al. (2018) or Smeed et al. (2018), the latter already used in the manuscript at a later point (line 423). Should be cited here as well.
  (l. 30) added AMOC strength RAPID references: Moat et al. (2018), Smeed et al. (2018), and Frajka-Williams et al. (2019).
- 6. ✓ 37 "0.8 Sv of relatively fresh Pacific" reference? For example Woodgate and Aagaard (2005) (1. 37) added the suggested reference for Bering Strait transport.
- 7. ✓ 39f "Freshwater is also exchanged with the Mediterranean Sea which is strongly evaporative." I would rather term this a salinity exchange, because Mediterranean outflow is very salty, i.e. the Atlantic provides "freshwater", which is in this terminology somewhat awkward.
  (1. 40) rewrote sentence incorporating the suggestion.
- 8.  $\checkmark$  42 "... and advect salt meridionally when there is a zonal salinity gradient." In the same sentence it is said that the gyres are wind-driven. This part sounds like they are driven by a zonal salinity gradient. I suggest to rephrase this part: "... and advect any zonal salinity gradient also in meridional direction." In context with the previous sentence, this most importantly means that the salinity differences caused by precipitation patterns in the ITCZ are advected poleward by the gyres. Maybe this should be stressed more.
  - (l. 42) reformulated the sentence as suggested.
- 9. ✓ 43 "under greenhouse gas increases" rather is "under increasing greenhouse gas concentrations" (1. 45) reformulated the sentence as suggested.
- 10. ✓ 58-66 very nice, brief explanation of the impact of freshwater import from the south on the salt-advection feedback. However, in principle the AMOC does not import freshwater to the North Atlantic but rather negative salinity anomalies (in models often a virtual salt flux anyways, see line 105). Also, a note on the calculation of FovS would be helpful, i.e. the typical reference salinity and whether zonally averaged velocities are used (AMOC streamfunction) or actual transports in 3-D are computed / is there a standard in place already?

(l. 60) We now mention the reference salinity in the introduction and refer to the appendix for the details of the calculations.

- 11.  $\checkmark$  117 How does the difference in vertical ocean grid resolution (42 vs. 60 levels) affect overflows in the North Atlantic? In particular resolution at Denmark Strait has the potential to significantly affect the AMOC.
  - (l. 132) additional text now clarifies this:

The 0.1° POP2 model grid has 42 levels to 6000 m while the LR-POP2 grid has 60 levels to 5500 m. In contrast to the HR ocean grid with its partial bottom cells and explicitely resolved overflows, the LR-CESM grid is defined with complete bottom cells and uses overflow parametrizations, e.g. between the Nordic Seas and the Atlantic (Smith et al., 2010). In the 0.1° POP2 model, the explicitely modeled Nordic Seas overflows compare favourably to observations (Ypma et al., 2019). The Mediterranean Outflow is not parameterized in the 1° POP grid but is modeled with a widened Strait of Gibraltar. Ultimately, the effect of the different vertical resolution is hard to disentangle as the horizontal mixing is represented very differently.

- 12. ✓ 132 more precise: "Green lines in Figure 1b,c mark the bounding latitudes which ..."
  (l. 154) specifically mentioning green lines now.
- 13. ✓ 165 Meaning of this introductory sentence not quite clear. Circulation changes between models must affect much more than just Bering Strait exchange and Mediterranean outflow. Maybe simply drop this sentence? Or move to line 199.
   (1. 188) rewrote this introductory sentence.
- 14. ✓ 174f please provide depth in meters (not km)!
  (ll. 194, 247 and others) changed all depth units to meters.
- 15.  $\checkmark$  174 The reference should more clearly point to Figure 3d for the northward extent of AAIW, which by the way seems not to reach 20°N (maybe 10°N) as stated here, and Figure 3j for the low salinity signature of AAIW.
  - (l. 194) changed description as suggested.
- 16. ✓ 176 Figure 3i shows a section at 34°S and thus cannot serve as a reference for the addressed bias at 15-30°N.
  (1, 100)
  - (l. 198) removed reference to Fig. 3i.
- 17. ✓ 191 a/dd "modelled" in "... this is the salinity of modelled North Atlantic ..."
  (l. 215) changed as suggested.
- 18. ✓ 227 I assume brine rejection is counted as negative freshwater flux into the ocean; add parentheses: "... sea ice melt (and brine rejection) ... defined as positive (negative) freshwater fluxes ..." (l. 255) made suggested changes.
- 19. 244, 284ff and Appendix B: It is not quite clear to me whether your method of computing eddy transports accounts for eddy induced velocities from the GM parameterization, which I believe is used in the LOW model run. While it is quite obvious that the velocity field of LOW is much smoother than the one in HIGH (as you point out for Figures 5d and 5e), the unresolved eddy fluxes should partly be compensated by the eddy mixing scheme (GM as noted only later in line 289), which would provide eddy induced velocities. These should be included in the eddy transport discussed with Figure 7. In this respect, the comment on line 280f should be moved upward and included in the introduction to section 3.3. [I was a bit impatient when reading this page and would have preferred to read the discussion of lines 280ff earlier. However, when reading this page again, I now think the structure is OK only that a small comment in line 259 would help, such as "In the following we take a closer look at these two differences."

(l. 290) added the note as suggested.

(App. B) The GM induced velocities are significantly smaller than the explicitely calculated velocities (Nooteboom et al., 2020). The associated salt and freshwater transport is very small compared to the advection terms by the explicitely calculated velocities. We will add a Figure in Appendix B to show this.

20. 248 Providing the year 2100 value of the linear trend seems a good way to limit the effect of internal variability in illustrating the changes under RCP scenario. However, it would be helpful to also note the correlation and significance of the linear trends in the text (or a table?) as a goodness-of-fit affirmation.

[how to best implement this?] appendix figure with R/p-values vs. latitude for HR/LR-CESM

21.  $\checkmark$  257 The obvious difference between the green CTRL lines ( $F_{eddy}$ ) of HIGH and LOW runs should

be pointed out first before discussing the deviations in trends, i.e. that LOW does not "exhibit the negative transport trend" at the SPG-STG boundary.

Each paragraph describes the CTRL mean state and RCP trend of one freshwater transport component. The second paragraph of the 'Meridional transport of freshwater' subsection is concerned with the total transport, while the paragraph starting at line (l. ??) described the eddy term. In the original manuscript we introduced Fig. 9 only in the second paragraph before the total transport which may have obscured this structure. We now introduce the figure in the first paragraph and mention the concerned term in the beginning of each paragraph.

- 22. ✓ 272 here or earlier: reference Yang et al. (2020) for shift in ocean gyres [see full reference below].
  (ll. 211, 392) mentioned that the shift is expected under climate change and inserted the reference.
- 23. ✓ 275 It could be noted in addition to the present discussion that the differences between LOW and HIGH are not only due to resolving eddies but also due to a generally better representation of boundary currents (azonal flow system) in HIGH, which I assume is the case.
  (1. 304) mentioning this point now.
- 24. ✓ 280f this statement should be made earlier (see comments on line 244 and App.B).
  (ll. 287, ??) as explained in above (line 257 comment), we changed the structure of the text to resolve the issue.
- 25. ✓ 343 It would be helpful if you indicate the bar color you are referring to. For the 32% vs 18% increase I assume you mean the total, i.e. red bars. This is a very nice, detailed analysis. And I also like Figure 9 very much but it is somehow difficult to keep track of the bars (colors) each sentence refers to. Adding hints for the color would help to link text and figure.
  (1. 379 and whole 'Freshwater budget' subsection) We added more color hints throughout the subsection and rephrased some sentences to facilitate the understanding of the plot. The percentage increases actually referred to the total surface freshwater flux which has now been clarified in the text.
- 26. ✓ 345 If sea ice is one of the few bigger differences between HIGH and LOW worth noting, then please add this flux to Figure 9! A "(not shown)" is not very satisfying here.
  (1. 381) we recalculated the Arctic surface fluxes and focus now on the much stronger enhancement of the hydrological cycle of HR-CESM compared to LR-CESM; sea ice export and import only plays a minor role and we do not mention it anymore.
- 27. ✓ 363 I cannot see the advantage of presenting the spin up timeseries for this discussion. (see more comments on Figure 10)
   See comment for Fig. 10 below.
- 28. ✓ 391 one reference to Jüling et al. (2020) is sufficient in this line. Also, this citation lacks a journal and DOI in the reference list. Is the paper accepted/published already?
  (l. 434) removed the second reference. Jüling et al., 2020 is currently in the review process of Ocean Science and is thus accessible (DOI:10.5194/os-2020-85; see reference below).

**Tables**

1. ✓ Table 1 In addition to the start year of the RCP run, please provide the length of the CTRL run (spinup?). Otherwise it looks like the high-resolution model was only spun up for 200 years, which

would be very short to study the AMOC. Did these runs branch off of any longer CESM spinup? (l. 118) we added a description of how the simulations were initialized.

2. ✓ Table 2 I recommend to use sign + for transports into the Atlantic-Arctic basin. (I would think the Mediterranean provides a net (virtual) salt inflow to the system, i.e. +1.2 and -0.032.) (Tbl. 2) changed sign as suggested and updated table caption.

**Figures**

- ✓ Figure 1: please use fewer colors for all plots to enhances visibility of actual differences.
   (Fig. 1) reduced the color increments to 1 K (K/century) for all panels.
- 2. ✓ Figure 5: the offset applied to the linear trend lines is not necessary, I think. Just use tone down the color a little bit, then it can be placed right in top of the smoothed timeseries without information loss.
  (Fig. 5) removed linear fit offset and changed color to grey.
- 3. ✓ Figure 7: thick and thin lines of CTRL and RCP are barely distinguishable. Please increase the difference in thickness or use dashed lines for RCP.
   (Fig. 7) changed RCP to dashed linestyle.
- 4. ✓ Figure 10: I depreciate the change in timescale. I think this gives a wrong impression on the trend under RCP w.r.t. the equilibration process. Also, I cannot grasp the purpose of showing the spinup period at all. Why do you not focus on the last 100 years in both cases? Further, I suggest to add labels for the regimes defined by the sign on the y-axis, e.g. on the righthand side y-axis. (Fig. 10) the time scale change has been removed and the values of both simulations are now plotted in a single planel. The regimes are labelled now on the right. The Figure caption and references to the figure in the text have been updated accordingly.

We still show all available data including the spin-up of the control simulation to make the point (l. 492) that the negative  $F_{ovS}/\Sigma$  sign of the HR-CESM simulation before 150 years is an artifact of it not being equilibrated. This is depite no trend being apparent in the first 100 years.

5. ✓ Appendix B: Does the velocity v in your calculations for the LOW model include eddy induced velocity components from an eddy parameterization such as GM? I think this should be included for a fair comparison between LOW and HIGH. The salinity distribution will inherently include the effect of such parameterization but does v?

See answer to comment 19 for a discussion of the GM term.

6. 511: Why annual mean? Since you have mean(vS) from model output you can compute the eddy transport also on monthly basis and thus eliminate the effect of seasonal variability from your calculation.

(ll. ??, 319, 555) calculated the eddy term now with the monthly output fields so as to avoid seasonal effects as suggested. We also adapted the text to reflect these changes.

**References**

Behrens, Erik, Arne Biastoch, and Claus W. Böning (2013). "Spurious AMOC trends in global ocean sea-ice models related to subarctic freshwater forcing". In: *Ocean Modelling* 69, pp. 39–49. ISSN:

14635003. DOI: 10.1016/j.ocemod.2013.05.004. URL: http://dx.doi.org/10.1016/j.ocemod. 2013.05.004.

- Frajka-Williams, Eleanor et al. (2019). "Atlantic meridional overturning circulation: Observed transport and variability". In: *Frontiers in Marine Science* 6.JUN, pp. 1–18. ISSN: 22967745. DOI: 10.3389/ fmars.2019.00260.
- Jüling, André, Anna Von Der Heydt, and Henk A Dijkstra (2020). "Effects of strongly eddying oceans on multidecadal climate variability in the Community Earth System Model". In: Ocean Science Discussions, pp. 1–24. DOI: 10.5194/os-2020-85.
- Moat, B.I. et al. (2018). Atlantic meridional overturning circulation observed by the RAPID-MOCHA-WBTS (RAPID-Meridional Overturning Circulation and Heatflux Array-Western Boundary Time Series) array at 26N from 2004 to 2018 (v2018.2). DOI: 10/d3z4.
- Nooteboom, Peter D. et al. (2020). "Resolution dependency of sinking Lagrangian particles in ocean general circulation models". In: *PLOS ONE* 15.9, e0238650. ISSN: 1932-6203. DOI: 10.1371/journal. pone.0238650. URL: https://dx.plos.org/10.1371/journal.pone.0238650.
- Peltier, W. Richard and Guido Vettoretti (2014). "Dansgaard-Oeschger oscillations predicted in a comprehensive model of glacial climate: A "kicked" salt oscillator in the Atlantic". In: *Geophysical Research Letters* 41.20, pp. 7306–7313. ISSN: 19448007. DOI: 10.1002/2014GL061413.
- Smeed, D. A. et al. (2018). "The North Atlantic Ocean Is in a State of Reduced Overturning". In: *Geophysical Research Letters* 45.3, pp. 1527–1533. ISSN: 19448007. DOI: 10.1002/2017GL076350.
- Smith, R. et al. (2010). The Parallel Ocean Program (POP) Reference Manual. Tech. rep.
- Woodgate, Rebecca A. and Knut Aagaard (2005). "Revising the Bering Strait freshwater flux into the Arctic Ocean". In: *Geophysical Research Letters* 32.2, pp. 1–4. ISSN: 00948276. DOI: 10.1029/2004GL021747.
- Yang, Hu et al. (2020). "Poleward Shift of the Major Ocean Gyres Detected in a Warming Climate".
   In: Geophysical Research Letters 47.5. ISSN: 19448007. DOI: 10.1029/2019GL085868.
- Ypma, S.L. et al. (2019). "Pathways and watermass transformation of Atlantic Water entering the Nordic Seas through Denmark Strait in two high resolution ocean models". In: *Deep Sea Research Part I: Oceanographic Research Papers* 145.August 2018, pp. 59–72. ISSN: 09670637. DOI: 10.1016/ j.dsr.2019.02.002.

---

## Author Comment (AC1)

DOI: 10.5194/os-2020-76
Version: Revision
Title: The Atlantic's Freshwater Budget under Climate Change in the Community Earth System Model
Authors: André Jüling, Xun Zhang, Daniele Castellana, Anna S. von der Heydt, Henk A. Dijkstra

Point by point reply to reviewer #1

November 27, 2020

We thank the reviewer for their careful reading and for the useful comments on the manuscript. We marked the points we have already addressed with ticks.

**1 Reviewer Summary:**

This manuscript gives a very detailed description of the components of the Atlantic Freshwater budget and how it changes under future climate projections. The study uses two versions of the CESM with different resolutions, one which is strongly eddying and the other which parametrizes eddies in the ocean. The results show that the higher resolution model has smaller biases than the lower resolution model. The authors also show that there isn't much difference in the response of the AMOC to the future CO2 projections between the high and low resolution simulations. While there are aspects of the freshwater budget between the high and low resolution simulations that similar the high resolution simulation has the additional transport due to eddies at the gyre boundaries. I feel that the detail of the freshwater budget presented in this study is of interest to the scientific community. However, while I feel that the scientific content is sound and contents is well organized, I believe there is quite a bit of room for improvement to this study.

**2 Major Comments:**

1. Throughout the entire study references to existing literature is a bit on the sparse side. For instance, the second paragraph of the introduction only has one reference. References where values based on observations/models are stated should be included (e.g. line 33 – Frajka-Williams et al. (2019) (or Smeed et al. (2018)) for AMOC strength or line 37 – Woodgate and Aagaard (2005) for Bering Strait through flow). Another example is when discussing the freshwater budget comparing how the results in the paper support or contradict other previous studies (i.e. Skliris et al. (2020) in their figure 11 do something very comparable to the manuscript's figure 9, Similarly, the studies Yin and Stouffer (2007) and Mecking et al. (2016) also do freshwater budgets).

(1.30) added Moat et al. (2018), Smeed et al. (2018), and Frajka-Williams et al. (2019) references to AMOC strength.

(1.37) added Woodgate and Aagaard (2005) reference to Bering Strait transport.

We will methodically go through the manuscript and check for missing references to previous work.

2.  $\checkmark$  A major difference between the high and low resolution simulations is the model's ability to handle eddies. Therefore, it would be nice to see a figure showing the differences in the eddy activity between the HIGH and LOW models (e.g. something similar to Delworth et al. (2012)

their figure 14).

(Figs. 4 and A4, l. 149) added and Atlantic and global map of the standard deviation of the observed sea surface height and the model dynamic sea level as in Brunnabend et al. (2017).

- 3.  $\checkmark$  The short names HIGH and LOW used through the manuscript when they aren't followed by the word simulation makes a quick read of some sentences confusing (e.g. line 158, 178, 213, etc). Throughout the manuscript we changed the names of HIGH and LOW to HR-CESM and LR-CESM, respectively.
- 4.  $\checkmark$  It would be very helpful to have some of the figures include extra panels (redone to show anomalies (differences between CTRL and RCP) as opposed to just the absolute values. In particular, this would be very helpful for Figures 7 and 9, making it a lot clearer what the changes in the future climate projections are. The text would probably benefit from a few small tweaks to reflect these figures this as well.

(Fig. 7) added panels to show the linear trends. We further removed horizontal grid lines and and legends in (b) to focus attention on the data.

(Fig. 9) added a panel to show only the trends. For further changes see comment below.

5. For several reasons I find Figure 9 nice but at the same time too complex. I really like comparisons of budgets using bar charts since the make it easy to see the relative differences between the different components. Panel a is good but panel b contains too much information. There is a lot of information on panel b, perhaps breaking panel b down into several panels will make it simpler, e.q. have a panel of surface flux breakdown and another for advection break down. The axis on the bottom of panel b are also confusing, first of all that they are different sizes for horizontal and vertical and where do the units, Sv/100yr come into play? Also, the arrows on the end of the bars are difficult to see differences between the HIGH and LOW simulations. It would be helpful to make an anomaly version of this figure. (Fig. 9) We will update this figure.

**3 Minor Comments:**

- 1.  $\checkmark$  At a few points in the manuscript a freshwater and salinity budget is mentioned, but only the freshwater budget is discussed. Even though they are quite similar it is probably worth just mentioning freshwater budget. (e.g. line 1 and other places) (ll. 1, ??, 357,494) removed mention of 'salinity budget'.
- 2.  $\checkmark$  Line 26 Mecking et al. (2016) also showed this (1.25) reference added
- 3.  $\checkmark$  Lines 32 and 37 Do you mean approximately instead of some? (ll. 30, 37) replaced 'some' with 'approximately'.
- 4.  $\checkmark$  Line  $\frac{44}{45}$  What about heat flux changes? i.e. Gregory et al., 2005 (1.47) We now mention the results of Gregory et al. (2005).
- 5. ✓ Lines 51-66 The references Weaver et al. (2012) and Liu et al., 2014 are also quite relevant for this paragraph.

(1.66) added these references in the appropriate position.

- 6. ✓ Line 66 reference? Is this line even needed?
  (1.70) we added Drijfhout et al. (2011) as a reference for CMIP3 models and Weaver et al. (2012) for CMIP5 models to mention the sign of the two CMIP generations under increasing radiative forcing.
- 7.  $\checkmark$  Line 90 You are comparing ocean-only (Deshayes et al., 2013) to coupled simulations (Mecking et al., 2017). The ocean only simulations use salinity restoring which is potentially the reason why the ocean only simulations have a negative  $F_{ovs}$  as opposed to the difference in resolution. (1.93) clarified that Deshayes et al. (2013) uses an ocean only setup while Mecking et al. (2017) analyzes coupled models.
- 8. ✓ Line 113/114 What impact does only using CO2 have?
  (l.124) We added the following clarification to the description of the RCP scenario:

In 2100, the radiative forcing of  $CO_2$  alone is  $6.9 \text{ W m}^{-2}$ , or 80% of the  $8.5 \text{ W m}^{-2}$  of the RCP8.5 scenario (Vuuren et al., 2011). Not prescribing land use changes, has no effect on the global mean surface temperaturature in the RCP8.5 scenario (Davies-Barnard et al., 2014). Compared to the mean warming in 2100 of the two RCP8.5 CESM1/CAM5 simulations submitted to CMIP5 at  $4.4 \,^{\circ}$ C (Meehl et al. (2013); time series available at https://climexp.knmi.nl/CMIP5/Tglobal/), our LOW RCP simulation warmed only  $2.9 \,^{\circ}$ C, or 66% of the RCP8.5 value. The reduced warming until 2100 is both because of the aforementioned reduced radiative forcing, but also the fact that our simulation started from a nearly equilibrated, and hence relatively warm, year 2000 control simulation.

- 9. ✓ Line 139 reference for HadISST missing
  (1.160) added Rayner et al. (2003) as the HadISST reference.
- 10. ✓ Line 141-142 Is there a warming trend in the beginning of CTRL before the reference period for this study is taken? Since the reference period of HadISST is 10 years before 2000 and 20 years after, I would naively not expect a warming.

The surface climate of the CTRL simulation adjusts strongly until year 50 (as seen in the GMST), after this there is a very small warming trend (much smaller than the RCP GMST trend) as the deep ocean continues to equilibrate and a very small radiative imbalance remains. As mentioned in the text, since the CTRL simulation is nearly equilibrated it is expected to be warmer than a transiently warming historical climate (even if the 30 year observed climatology used here is actually centered around the year 2005).

11.  $\checkmark$  In some 2D figures there is a line of missing data for the LOW simulations (e.g. Figures 1,3c,i,4b,6).

(Figs. 1,2,3,4,6,8) fixed this cartopy issue in all maps.

- 12. ✓ Line 148 Reference to indicate that the warming hole response is expected i.e. Drijfhout et al. (2012)
  (1170) Ibb land to see the total of the basis of the second second
  - $\left( l.170\right)$  added sentence about the 'warming hole' and the suggested reference.
- Line 162/Figure 2e The integration of the surface fresh water fluxes is typically done North to South because that's the direction of the barotropic flow through the Atlantic (i.e. Skliris et al. (2020) Figure 6, Mecking et al. (2017) Figure 6), this way it will line up to the difference between inflow into the Bering Strait and outflow at 34S

(l.185) We only integrate P-E, not P-E+R as the mentioned studies, to compare with the P-E

ERA-Interim data. This data does not include a runoff mask that would enable us to compare this term as well.

- 14. ✓ Table 2, it would be nice to also include the transports at 34°S (or as close to it as possible, see Bryden et al. (2011) for an observational estimate).
  (Tbl. 2) appended 24°S transports to the table.
- 15. ✓ Table 2 how is salt transport defined and how does it relate to freshwater transport? (Tbl. 2) added the definition to the table caption.
- 16.  $\checkmark$  Line 180 I don't understand the barotropic streamfunction computation, the constant of integration should have an x/longitude dependence and not along the coast of the Atlantic side of Africa

(1.202) In the calculation of the streamfunction the coasts of islands and continents have constant values implying no flow across these coasts. As we focus here on the Atlantic with only few and small islands, we display the streamfunction referenced to the African coast streamfunction value. In the HR-CESM simulation, the barotropic streamfunction is diagnosed and written out as ocean model output. Our approximation agrees well with this model output field. We chose to present a consistent estimate for both models.

- 17. ✓ Line 211 By colours do you mean shading?
  (1.240) changed wording as suggested.
- 18. ✓ Line 221 Reference for AMOC strength
   (1.249) added Frajka-Williams et al. (2019) reference for AMOC strength
- 19. ✓ Line 229 The freshwater loss seems large (i.e. larger than CMIP5, see Skliris et al. (2020) Figure 6)
  (1.258) we checked the calcuation again and this is the total surface flux, we noted that this is larger than the CMIP5 ensemble mean freshwater loss.
- 20. ✓ Line 234 Which linear trends are being referred to here?
   (1.264) clarified in the text that these linear trend values referred to the total surface freshwater flux.
- 21. Figure 7 why is 36N masked?(Fig. 7) We will fix this.
- 22. Figure 7 Do freshwater transports through the Bering Strait and Strait of Gibraltar change? (Fig. 7) They change very little compared to other changes, the trends will be included in a new version of Figures 7 and 9.
- 23. ✓ Figure 7 and 10, most obvious in Figure 7, the thick and thin line thicknesses are very difficult to distinguish.
  (Figs. 7, 10) changed RCP to dashed linestyle.
- 24. Line 256 (some other places later) it would be nice to reference earlier figures/figure panels at the end of sentences to help make connections i.e. line 256 –Fig. 6k/l and line 266 Fig. 7 We will add references to the figure panels.
- 25. Figure 8 Is it worth including 34S section and/or Atlantic zonal mean?
  We will plot the section salinity trends at 34°S similar to Fig. 3. The zonal mean may not add much information as azonal trends as seen in Fi. 8 are averaged out.

26. ✓ Figure 8 – the freshening trend in the SPG is in line with the warming hole that goes along with a weakening AMOC (i.e. Menary and Wood (2018) Fig. 7) – should be mentioned in the text below

(1.331) added a sentence describing this and referening the suggested study.

- 27.  $\checkmark$  Line 305 Any idea why there are these differences in salinity? We abstain from speculating as this requires a more in-depth study of the local salinity budget which is outside the focus of this study.
- 28. ✓ Line 321 'durface' should be 'surface'
  (1.353) fixed typo
- 29. ✓ Line 361 Also in Liu et al. (2014)
  (1.400) citation added

**References**

- Brunnabend, S. E. et al. (2017). "Changes in extreme regional sea level under global warming". In: Ocean Science 13.1, pp. 47–60. ISSN: 18120792. DOI: 10.5194/os-13-47-2017.
- Bryden, Harry L., Brian A. King, and Gerard D. McCarthy (2011). "South Atlantic overturning circulation at 24S". In: Journal of Marine Research 69.1, pp. 39–56. ISSN: 15439542. DOI: 10.1357/ 002224011798147633.
- Davies-Barnard, T. et al. (2014). "Full effects of land use change in the representative concentration pathways". In: *Environmental Research Letters* 9.11. ISSN: 17489326. DOI: 10.1088/1748-9326/9/11/114014.
- Delworth, Thomas L. et al. (2012). "Simulated climate and climate change in the GFDL CM2.5 high-resolution coupled climate model". In: *Journal of Climate* 25.8, pp. 2755–2781. ISSN: 08948755. DOI: 10.1175/JCLI-D-11-00316.1.
- Deshayes, J. et al. (2013). "Oceanic hindcast simulations at high resolution suggest that the Atlantic MOC is bistable". In: *Geophysical Research Letters* 40.12, pp. 3069–3073. ISSN: 00948276. DOI: 10. 1002/grl.50534. URL: http://doi.wiley.com/10.1002/grl.50534.
- Drijfhout, Sybren, Geert Jan van Oldenborgh, and Andrea Cimatoribus (2012). "Is a decline of AMOC causing the warming hole above the North Atlantic in observed and modeled warming patterns?" In: Journal of Climate 25.24, pp. 8373–8379. ISSN: 08948755. DOI: 10.1175/JCLI-D-12-00490.1.
- Drijfhout, Sybren S., Susanne L. Weber, and Eric van der Swaluw (2011). "The stability of the MOC as diagnosed from model projections for pre-industrial, present and future climates". In: *Climate Dynamics* 37.7-8, pp. 1575–1586. ISSN: 09307575. DOI: 10.1007/s00382-010-0930-z.
- Frajka-Williams, Eleanor et al. (2019). "Atlantic meridional overturning circulation: Observed transport and variability". In: Frontiers in Marine Science 6.JUN, pp. 1–18. ISSN: 22967745. DOI: 10.3389/ fmars.2019.00260.
- Gregory, J. M. et al. (2005). "A model intercomparison of changes in the Atlantic thermohaline circulation in response to increasing atmospheric CO2 concentration". In: *Geophysical Research Letters* 32.12, pp. 1–5. ISSN: 00948276. DOI: 10.1029/2005GL023209.
- Liu, Wei, Zhengyu Liu, and Esther C. Brady (2014). "Why is the AMOC monostable in coupled general circulation models?" In: *Journal of Climate* 27.6, pp. 2427–2443. ISSN: 08948755. DOI: 10.1175/JCLI-D-13-00264.1.
- Mecking, J. V. et al. (2016). "Stable AMOC off state in an eddy-permitting coupled climate model". In: *Climate Dynamics* 47.7-8, pp. 2455–2470. ISSN: 14320894. DOI: 10.1007/s00382-016-2975-0.

- Mecking, J.V. et al. (2017). "The effect of model bias on Atlantic freshwater transport and implications for AMOC bi-stability". In: *Tellus A: Dynamic Meteorology and Oceanography* 69.1, p. 1299910. ISSN: 1600-0870. DOI: 10.1080/16000870.2017.1299910.
- Meehl, Gerald A. et al. (2013). "Climate change projections in CESM1(CAM5) compared to CCSM4". In: Journal of Climate 26.17, pp. 6287–6308. ISSN: 08948755. DOI: 10.1175/JCLI-D-12-00572.1.
- Menary, Matthew B. and Richard A. Wood (2018). "An anatomy of the projected North Atlantic warming hole in CMIP5 models". In: *Climate Dynamics* 50.7-8, pp. 3063–3080. ISSN: 14320894. DOI: 10.1007/s00382-017-3793-8.
- Moat, B.I. et al. (2018). Atlantic meridional overturning circulation observed by the RAPID-MOCHA-WBTS (RAPID-Meridional Overturning Circulation and Heatflux Array-Western Boundary Time Series) array at 26N from 2004 to 2018 (v2018.2). DOI: 10/d3z4.
- Rayner, N. A. et al. (2003). "Global analyses of sea surface temperature, sea ice, and night marine air temperature since the late nineteenth century". In: *Journal of Geophysical Research* 108.D14, p. 4407. ISSN: 00113891. DOI: 10.1029/2002JD002670.
- Skliris, Nikolaos et al. (2020). "Changing water cycle and freshwater transports in the Atlantic Ocean in observations and CMIP5 models". In: *Climate Dynamics* 19.0123456789, pp. 2017–8976. ISSN: 0930-7575. DOI: 10.1007/s00382-020-05261-y.
- Smeed, D. A. et al. (2018). "The North Atlantic Ocean Is in a State of Reduced Overturning". In: *Geophysical Research Letters* 45.3, pp. 1527–1533. ISSN: 19448007. DOI: 10.1002/2017GL076350.
- Vuuren, Detlef P. van et al. (2011). "The representative concentration pathways: An overview". In: *Climatic Change* 109.1, pp. 5–31. ISSN: 01650009. DOI: 10.1007/s10584-011-0148-z.
- Weaver, Andrew J. et al. (2012). "Stability of the Atlantic meridional overturning circulation: A model intercomparison". In: *Geophysical Research Letters* 39.20, pp. 1–8. ISSN: 00948276. DOI: 10.1029/2012GL053763.
- Woodgate, Rebecca A. and Knut Aagaard (2005). "Revising the Bering Strait freshwater flux into the Arctic Ocean". In: *Geophysical Research Letters* 32.2, pp. 1–4. ISSN: 00948276. DOI: 10.1029/2004GL021747.
- Yin, Jianjun and Ronald J. Stouffer (2007). "Comparison of the stability of the Atlantic thermohaline circulation in two coupled atmosphere - Ocean general circulation models". In: *Journal of Climate* 20.17, pp. 4293–4315. ISSN: 08948755. DOI: 10.1175/JCLI4256.1.